# ALIGN AFTER PRE-TRAIN: IMPROVING MULTILINGUAL GENERATIVE MODELS WITH CROSS-LINGUAL ALIGNMENT

## ABSTRACT

Multilingual generative models obtain remarkable cross-lingual capabilities through pre-training on large-scale corpora. However, they still exhibit a performance bias toward high-resource languages, and learn isolated distributions of sentence representations across languages. To bridge this gap, we propose a simple yet effective alignment framework exploiting pairs of translation sentences. It aligns the internal sentence representations across different languages via multilingual contrastive learning and aligns model outputs by answering prompts in different languages. Experimental results demonstrate that even with less than 0.1 ‰ of pre-training tokens, our alignment framework significantly boosts the cross-lingual abilities of generative models and mitigates the performance gap. Further analysis reveals that it results in a better internal multilingual representation distribution of multilingual models.

## 1 INTRODUCTION

Multilingual generative language models achieve impressive universality across many languages by pre-training on large-scale unsupervised multilingual corpora (Liu et al., 2020; Xue et al., 2021; Lin et al., 2022; Scao et al., 2022; Soltan et al., 2022; OpenAI, 2022). However, models still show a strong language bias toward high-resource languages (Asai et al., 2023), even the state-of-the-art multilingual generative models like GPT-4, exhibiting a 27.5% relative performance gap between English and Telugu in MMLU (OpenAI, 2023). This challenge partly arises from the significant linguistic resource imbalance among languages, which is hard to address solely through corpus scaling or balancing. Given such a model with language bias and the huge cost of re-training, how can we improve its cross-language capabilities and alleviate the language bias using limited data?

Through visualizing the sentence representations in the multilingual generative model by mean pooling, we find that there is a distinct gap between the sentence representation distributions for different languages like Figure 1(a) (the multilingual ones are shown in Appendix B.3). This is similar to learning representations for each language separately in the model, which is more challenging for multilingual models to transfer the knowledge learned from other languages. It is interesting to investigate whether the cross-lingual ability of multilingual generative models will be promoted by learning a better-aligned representation distribution.

To address the above issues, we propose a cross-lingual alignment framework named A̲lign a̲Fter P̲re-train (**AFP**), which aims to exploit translation pairs to narrow the gap between languages in the multilingual generation model. To be specific, our method can be divided into the following two modules: 1) **Multilingual Contrastive Learning (MCL)** on internal representations: we treat a pair of translation sentences between two languages as the positive example for contrastive learning, and pull the sentence representations in two languages to be closer within the multilingual generated model. This method intends to reduce the differences between languages from the internal representations of the model. 2) **Cross-lingual Instruction Tuning (CIT)** on the outputs: models must learn to answer in the target language given a prompt from the source language. It requires models to obtain a better cross-lingual understanding and generation ability.

After extensive experiments and evaluation, it can be found that AFP greatly improves the performance of multilingual generative models, including XGLM and BLOOM, in cross-lingual natural

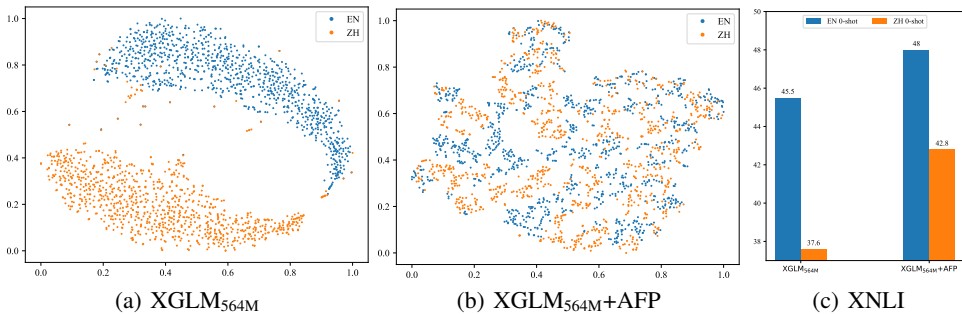

(a) XGLM$_{564M}$        (b) XGLM$_{564M}$+AFP        (c) XNLI

Figure 1: (a, b) Our method aligns the internal EN-ZH sentence representations of XGLM$_{564M}$, which are shown in t-SNE. (c) And it also mitigates the performance gap on XNLI.

language inference, multilingual reasoning, and other tasks using less than 1M parallel samples. The performance gap between languages is much mitigated, e.g., the relative performance gap reduces 6.53% in 0-shot performance of XNLI between English and Chinese (Figure 1(c)). Our method also advances the performance on unseen languages for models, e.g., the Chinese performance of Llama, which is pre-trained on the corpus mainly in English (Touvron et al., 2023a;b). Further analysis reveals that the alignment between languages in the model has been improved as illustrated in Figure 1(b) after training via AFP. In addition, experimental results show that the cross-lingual instruction tuning task improves cross-lingual capabilities more than the multilingual instruction tuning task with the same parallel samples.

To sum up, our contribution lies in two aspects:

- We propose a simple yet effective cross-lingual alignment framework named AFP, including the internal representation alignment (MCL) and output alignment (CIT), to exploit the parallel corpus. Quantitive analysis shows that the internal multilingual representation distributions have been improved after using AFP.

- Experimental results demonstrate that our method greatly improves the cross-lingual ability of generative models, including multilingual ones and models pre-trained on English corpus, by using less than 1M parallel samples. After alignment, models can also be applied with other methods to enhance performance further.

## 2   Aligning Multilingual Representations and Outputs of Generative Models

As shown in Figure 2, our framework AFP contains the following two modules: 1) Multilingual contrastive learning (Section 2.1), which aims to align the internal representations of models across different languages. 2) Cross-lingual Instruction Tuning (Section 2.2), which requires models to align the outputs between different languages.

### 2.1   Multilingual Contrastive Learning

To align the internal multilingual representation of models, we exploit the contrastive learning method, which is generally found effective in aligning the representations from different modalities in multi-modal work (Radford et al., 2021; Xu et al., 2021; Liang et al., 2022). Hence, translation pairs are regarded as positive instances with closely aligned semantics in multilingual contrastive learning, and we pull their internal representations closer. The other sentences in the same batch are taken as the negative samples for the translation pair.

Formally, to align the $l$-th layer of model $f(\theta)$, the sentence representations $(h_i, h_i^+)$ is calculated as follows:

$$h_i = g(f_l(s_i; \theta)), \; h_i^+ = g(f_l(s_i^+; \theta)) \tag{1}$$

where $f_l(\cdot)$ represents the output from the $l$-th layer, $g(\cdot)$ is the pooling method to obtain the sentence representation for decoder models, e.g., mean pooling or max pooling, and $(s_i, s_i^+)$ is a parallel sample from $\mathcal{D} = \{(s_1, s_1^+), ..., (s_n, s_n^+)\}$. We determine the specific layer to align according to

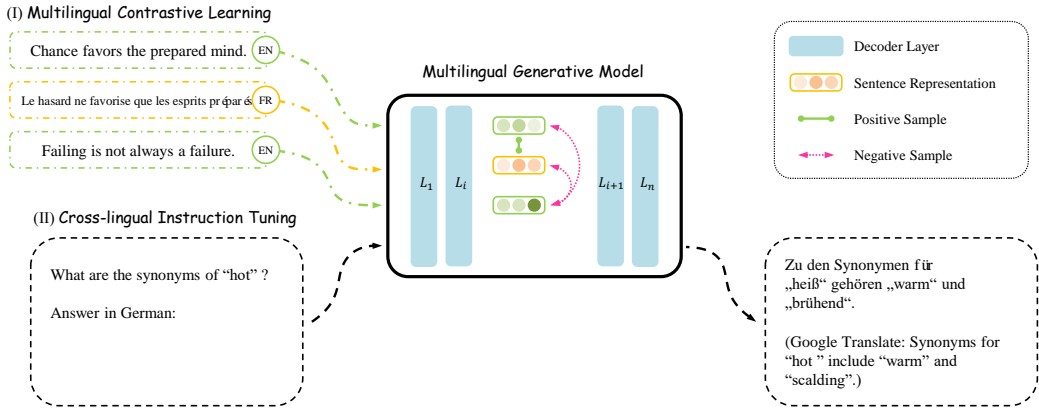

Figure 2: Illustration of how to align the internal representations and outputs of multilingual generative models in AFP. (I) Given a translation parallel sample as the positive sample, multilingual contrastive learning pulls their representations together and pushes apart the ones from other samples. (II) Multilingual generative models are required to answer in the target language to align the outputs across languages.

the performance of models on the dev set, and we find that the first layer after embedding comes to better performance (please refer to Section 3.2.2 for more details). Then, the training objective of Multilingual Contrastive Learning (MCL) is:

$$\mathcal{L}_{MCL}(\theta) = \mathop{\mathbb{E}}_{(s_i, s_i^+) \sim \mathcal{D}} \left[ -\log \left( \frac{e^{\text{sim}(h_i, h_i^+)/\tau}}{\sum_j e^{\text{sim}(h_i, h_j)/\tau}} \right) \right] \tag{2}$$

where $\text{sim}(\cdot)$ is used to determine the similarity between representations, which is cosine similarity in this work, $h_j$ is the sentence representation of $s_j$ in the mini-batch containing $(s_i, s_i^+)$, and $\tau$ is a temperature hyper-parameter.

## 2.2 CROSS-LINGUAL INSTRUCTION TUNING

To further align the output of multilingual generative models, we introduce a method named Cross-lingual Instruction Tuning (CIT), which imposes models to respond in the target language given the source language as the context. It is more difficult than the multilingual instruction tuning task, which prompts and answers in the same language for each sample, and requires a better cross-lingual understanding and generation ability for multilingual generative models.

Specifically, given a pair of context and response $(c_i^a, r_i^a)$ from a Dataset $\mathcal{D}^a$ in the same language $a$, e.g., an English instruction tuning dataset like FLAN or Alpaca (Wei et al., 2022; Wang et al., 2023; Taori et al., 2023), response $r_i^a$ is first translated into the target language $b$ by the translator $t^{a \to b}(\cdot)$. We append a prompt $p^b$ informing the target language $b$, e.g., "Answer in German" in Figure 2, at the end of context to construct the training sample $\left(c_i^{a \to b} = c_i^a + p^b, r_i^b = t^{a \to b}(r_i^a)\right)$ for CIT. Therefore, the loss function of CIT for the multilingual generative model $f(\theta)$ comes to:

$$\mathcal{L}_{CIT}(\theta) = \mathop{\mathbb{E}}_{(c_i^a, r_i^a) \sim \mathcal{D}^a} \left[ \sum_j -\log \left( \text{P}(r_{ij}^b | c_i^{a \to b}, r_{i, <j}^b; \theta) \right) \right] \tag{3}$$

where the target language $b$ has the possibility $p_{src} \in [0, 1]$ to be set the same as the source language $a$, which is a hyper-parameter and investigated in Section 3.2.3. When the target language is always the source language of the context ($p_{src} = 1$), it degenerates into the normal multilingual instruction finetuning method.

With the two modules of aligning methods mentioned before, Multilingual Contrastive Learning (MCL) and Cross-lingual Instruction Tuning (CIT), we obtain the following final loss function of our alignment framework AFP:

$$\mathcal{L}_{AFP}(\theta) = \mathcal{L}_{MCL}(\theta) + \alpha \mathcal{L}_{CIT}(\theta) \tag{4}$$

where $\alpha \in \mathbb{R}_0^+$ is a hyper-parameter to balance the two methods.

## 3 EXPERIMENTS

### 3.1 EXPERIMENTS SETTINGS

**Parallel Corpus** To cover more parallel samples from different domains and languages, we adopt a multilingual instruction tuning dataset named Bactrian-X (Li et al., 2023), which is translated into 52 languages from Alpaca (Taori et al., 2023) and Dolly (Conover et al., 2023) by Google Translate, and a multilingual machine translation dataset, OPUS-100 (Zhang et al., 2020), to align the models evaluated. Only 100k parallel samples are selected from OPUS-100 in our experiments to match the amount of Bactrian-X, which contains 67k samples for each language. The number of tokens used is about 20M, which is nearly 0.05 ‰ of tokens used in the pre-training of BLOOM (Scao et al., 2022).

**Language models** We apply AFP on two multilingual generative model structures, XGLM (Lin et al., 2022) and BLOOM (Scao et al., 2022), across three different parameter amounts. They are both pre-trained on large-scale unsupervised multilingual corpora with a more balanced sampling method across languages. Llama (Touvron et al., 2023a), which is mainly pre-trained on English corpus, is also included for comprehensive evaluation. Training settings and hyperparameters are reported in Appendix A.

**Multilingual Tasks** We evaluate the performance of models on the following benchmarks:

- **Natural Language Inference** We use XNLI (Conneau et al., 2018) in this task.
- **Paraphrase Detection** PAWS-X (Yang et al., 2019) is evaluated for this task.
- **Reasoning** We adopt XCOPA (Ponti et al., 2020), XStoryCloze (Lin et al., 2022) and XWinograd (Tikhonov & Ryabinin, 2021) in this task.
- **Machine Translation** For this task, we use FLORES-101 (Goyal et al., 2022).

The detailed descriptions and prompt formats for each task during evaluation are presented in Appendix C. We kept the same prompt formats across all multilingual generation models for a fair comparison.

### 3.2 BILINGUAL RESULTS AND ANALYSES

To make a comprehensive analysis of the influence on performance and representations in models, we first conduct bilingual alignment experiments in English and Chinese. Then we extended it to the multilingual alignment condition (Section 3.3).

Table 1 shows the experimental alignment results on EN-ZH parallel samples. These generative models, including three architectures with different amounts of parameters, are generally improved by our method. The average improvement is up to 3.31 using only 167k parallel samples, and the models with 7B parameters surpass the GPT-3 with comparable parameters after alignment. Specifically, models improve 4.28% on the first two natural language understanding tasks (XNLI and PAWS-X), and 2.67% on the other three reasoning tasks. After alignment using AFP, BLOOM shows a better performance than the BLOOMZ model with the same amount of parameters, which is fine-tuned on 78M multilingual instructions (Scao et al., 2022).

It is interesting to find that the model Llama pre-trained on mainly English corpus, also obtains improvement after bilingual alignment using AFP. The performance on the unseen language Chinese is even comparable with the one pre-training on an additional 20GB Chinese corpus (Cui et al., 2023). This result further proves the effectiveness of our method. We assume that this performance gain may benefit from better-aligned multilingual representations in models, which promotes the transfer of knowledge learned in the English corpus.

In addition to cross-lingual understanding and reasoning abilities, the multilingual generation ability of models has been improved. The bilingual translation results of XGLM models are reported in Table 2. Models not only obtain a better cross-lingual generation ability, but also show a more balanced generation performance than the vanilla ones between both directions. It is interesting to find that the average performance of models in the zero-shot condition improves from 0.1 to 4.2

Table 1: In-context learning results of models across different parameter scales on 5 datasets. The Average improvement is 3.31%, where 4.28% on the first two tasks and 2.67% on reasoning tasks. ‡ uses an additional 20GB Chinese corpus for pre-training. For a fair comparison, all results are obtained from the same in-context learning template illustrated in Appendix C.

| Model | XNLI EN-0/5 | XNLI ZH-0/5 | PAWS-X EN-0/5 | PAWS-X ZH-0/5 | XCOPA EN-0/5 | XCOPA ZH-0/5 | XStoryCloze EN-0/5 | XStoryCloze ZH-0/5 | XWinograd EN-0/5 | XWinograd ZH-0/5 | Avg |
|---|---|---|---|---|---|---|---|---|---|---|---|
| GPT-3$_{6.7B}$ | 55.3/52.8 | 42.4/45.9 | 60.6/59.7 | 53.2/54.1 | 73.6/74.5 | 55.0/57.7 | 73.6/74.5 | 55.9/54.5 | 64.6/68.1 | 71.5/72.2 | 61.0 |
| XGLM$_{564M}$ | 45.5/41.2 | 37.6/35.6 | 50.4/46.6 | 50.9/47.8 | 56.4/59.6 | 52.8/52.2 | 59.6/60.8 | 54.3/52.9 | 54.8/56.7 | 67.1/66.9 | 52.5 |
| +AFP | **48.1/46.5** | **41.6/42.5** | **54.2/53.8** | **53.2/52.8** | **62.0/62.2** | **59.0/58.8** | **62.2/62.5** | **56.3/56.1** | **55.6/59.0** | **67.5/67.3** | **56.1** |
| XGLM$_{1.7B}$ | 47.6/42.9 | 39.0/39.3 | 51.0/52.4 | 51.1/48.8 | 62.4/64.0 | 56.4/57.8 | 62.7/64.5 | 56.3/56.3 | 57.2/60.0 | 68.3/68.5 | 55.3 |
| +AFP | **48.3/46.9** | **43.1/43.7** | **55.2/55.7** | **54.5/53.8** | **66.8/68.8** | **61.4/61.6** | **65.5/66.7** | **59.3/59.6** | **58.8/64.7** | **68.7/69.6** | **58.6** |
| XGLM$_{7.5B}$ | 54.1/49.9 | 45.4/44.2 | 58.9/56.3 | 52.9/55.8 | 69.4/74.6 | 62.4/63.2 | 69.2/73.7 | 59.5/59.2 | 62.8/66.4 | 73.8/73.2 | 61.2 |
| +AFP | **55.0/54.7** | **48.0/48.8** | **64.8/61.2** | **57.8/56.4** | **72.2/75.6** | **64.4/66.8** | **72.0/74.7** | **62.7/63.4** | **65.2/68.2** | **75.8/74.0** | **64.1** |
| BLOOMZ$_{560M}$ | 43.8/44.5 | 41.5/40.7 | 52.4/51.2 | 54.1/52.9 | 54.8/57.2 | 52.0/52.8 | **61.2/61.7** | 56.4/55.0 | 54.8/55.4 | 62.3/65.1 | 53.5 |
| BLOOM$_{560M}$ | 44.4/40.4 | 41.1/40.3 | 50.5/52.3 | 49.0/49.4 | 53.0/57.4 | 49.8/54.0 | 55.2/58.2 | 57.9/53.2 | 54.3/55.6 | 63.9/64.9 | 52.2 |
| +AFP | **50.7/46.4** | **47.5/44.8** | **58.2/57.5** | **54.9/54.8** | **57.8/58.4** | **52.6/55.4** | 57.0/59.0 | **59.7/58.3** | **56.3/57.2** | **64.7/65.2** | **55.8** |
| BLOOMZ$_{1.7B}$ | 50.3/51.2 | 48.0/46.2 | 57.1/53.4 | 54.4/52.3 | 58.0/58.0 | 55.2/56.8 | 66.4/68.9 | 59.8/62.3 | 59.0/61.6 | 66.1/67.7 | 57.6 |
| BLOOM$_{1.7B}$ | 50.4/44.4 | 47.6/46.1 | 47.7/52.1 | 52.9/51.1 | 55.8/58.2 | 52.4/54.6 | 64.2/67.3 | 60.1/60.6 | 56.1/59.3 | 67.9/65.9 | 55.7 |
| +AFP | **52.9/51.3** | **49.8/48.8** | **61.0/58.0** | **56.9/56.0** | **60.8/61.6** | **55.4/58.2** | **66.4/69.0** | **63.3/63.3** | **59.3/60.7** | **68.3/66.1** | **59.4** |
| BLOOMZ$_{7.1B}$ | 51.1/52.0 | 49.7/48.0 | 63.6/62.2 | 56.9/56.1 | 61.2/62.4 | 57.6/59.8 | **73.7/76.9** | 62.1/63.9 | **64.1/66.9** | 66.1/68.5 | 61.1 |
| BLOOM$_{7.1B}$ | 54.0/48.7 | 48.1/47.5 | 59.9/60.4 | 53.2/51.4 | 58.0/58.8 | 54.0/54.8 | 70.4/73.5 | 64.3/64.8 | 60.6/63.8 | 71.4/67.7 | 59.3 |
| +AFP | **55.8/54.3** | **50.2/50.4** | **66.5/64.5** | **58.7/56.8** | **62.0/62.8** | **58.2/61.0** | 72.9/75.6 | **68.0/68.6** | 62.9/66.2 | **73.0/70.8** | **63.0** |
| Bactrian-X$_{7B}$ | 53.0/53.3 | 44.6/44.1 | 68.7/63.4 | 56.7/53.6 | 76.8/85.8 | 54.4/55.2 | 79.5/83.3 | 55.9/57.0 | 75.0/80.6 | 66.3/66.1 | 63.7 |
| ZH-Alpaca$_{7B}^{‡}$ | 51.7/52.9 | 47.2/46.2 | 67.6/62.8 | 57.2/54.8 | 73.2/83.8 | **57.6/60.8** | 76.6/79.3 | **57.4/58.3** | 71.4/74.8 | **67.9/68.5** | 63.0 |
| Llama$_{7B}$ | 54.5/49.0 | 45.9/44.9 | 67.8/64.2 | 55.4/53.1 | 74.6/84.2 | 55.8/57.4 | 77.0/80.7 | 55.0/55.5 | 72.3/79.4 | 66.1/65.5 | 62.9 |
| +AFP | **55.9/54.1** | 47.6/48.4 | **70.0/64.3** | **58.6/56.1** | **78.4/86.8** | 57.2/60.0 | **79.9/84.0** | 56.8/57.6 | **76.4/83.0** | 66.7/67.7 | **65.5** |

Table 2: Translation results on FLORES-101 devtest set.

| Model | EN→ZH 0 | EN→ZH 1 | EN→ZH 5 | EN→ZH 10 | ZH→EN 0 | ZH→EN 1 | ZH→EN 5 | ZH→EN 10 | Avg 0 | Avg 1 | Avg 5 | Avg 10 |
|---|---|---|---|---|---|---|---|---|---|---|---|---|
| XGLM$_{564M}$ | 0.0 | 1.0$_{\pm0.4}$ | 4.2$_{\pm0.5}$ | 4.2$_{\pm0.5}$ | 0.0 | 6.4$_{\pm0.6}$ | 8.1$_{\pm0.4}$ | 8.3$_{\pm0.5}$ | 0.0 | 3.7$_{\pm0.5}$ | 6.0$_{\pm0.4}$ | 6.3$_{\pm0.4}$ |
| +AFP | **2.0** | **4.2$_{\pm0.3}$** | **5.3$_{\pm0.3}$** | **5.9$_{\pm0.2}$** | **5.3** | **8.7$_{\pm0.6}$** | **9.0$_{\pm0.5}$** | **9.2$_{\pm0.4}$** | **3.7** | **6.5$_{\pm0.5}$** | **7.2$_{\pm0.3}$** | **7.6$_{\pm0.3}$** |
| XGLM$_{7.5B}$ | 0.0 | 13.6$_{\pm1.1}$ | 13.8$_{\pm0.9}$ | 13.9$_{\pm0.7}$ | 0.1 | 19.2$_{\pm0.7}$ | 19.5$_{\pm0.8}$ | 20.1$_{\pm0.4}$ | 0.1 | 16.4$_{\pm0.8}$ | 16.7$_{\pm0.6}$ | 17.0$_{\pm0.4}$ |
| +AFP | **2.5** | **14.5$_{\pm0.8}$** | **14.8$_{\pm0.6}$** | **15.0$_{\pm0.6}$** | **6.7** | **19.5$_{\pm0.6}$** | **19.6$_{\pm0.5}$** | **20.3$_{\pm0.3}$** | **4.6** | **17.0$_{\pm0.6}$** | **17.2$_{\pm0.5}$** | **17.7$_{\pm0.4}$** |

BLEU on average, which may come from the fact that the target language format of prompt used in cross-lingual instruction tuning is similar to the one in the machine translation task.

### 3.2.1 AFP BRINGS BETTER BILINGUAL REPRESENTATIONS

**Visualization of sentence representations.** Given 1k EN-ZH translation parallel samples, we visualize the sentence representations of XGLM$_{564M}$ and BLOOM$_{560M}$, which are obtained by the mean pooling method using the representations for each token in one sentence. In the vanilla models, there is a distinct separation between sentence representations from different languages (Figure 1(a) and 3(a)). However, the ones after alignment using AFP come to be more aligned and uniform (Figure 1(b) and 3(b)), which means our method promotes the representation of the model to be better-aligned from a qualitative point of view.

**Alignment and uniformity.** The distribution of multilingual representations is quantified by the two metrics, **alignment** and **uniformity** proposed by Wang & Isola (2020), for further analysis. Specifically, the alignment score measures the expected distance between the representations of positive samples, which are translation parallel samples for multilingual generative models, and is calculated as follows:

$$\mathcal{L}_{\text{align}} \triangleq \underset{(x,x^+)\sim\mathcal{D}_{pos}}{\mathbb{E}} \left\| f(x) - f(x^+) \right\|^2 \tag{5}$$

where $\mathcal{D}_{pos}$ is the distribution of positive samples.

In contrast, uniformity reflects the degree of uniformly distributed for representations:

$$\mathcal{L}_{\text{uniform}} \triangleq \log \mathop{\mathbb{E}}_{x,y \overset{i.i.d.}{\sim} \mathcal{D}} e^{-2\|f(x)-f(y)\|^2} \qquad (6)$$

where $x$ and $y$ are randomly sampled from the distribution $\mathcal{D}$. Therefore, The smaller $\mathcal{L}_{\text{align}}$ and $\mathcal{L}_{\text{uniform}}$ are, the better representations models learn.

Figure 3(c) illustrates the deviation of $\mathcal{L}_{\text{align}}$ and $\mathcal{L}_{\text{uniform}}$ for XGLM$_{564M}$ using different training methods on the same training data. The initial 5000 steps are visualized, one point for every 500 steps. We can find that the metrics are both decreasing using AFP, while the bilingual pre-training only improves the uniformity of representations. The results further prove that our method improves the multilingual representations within the multilingual generative models.

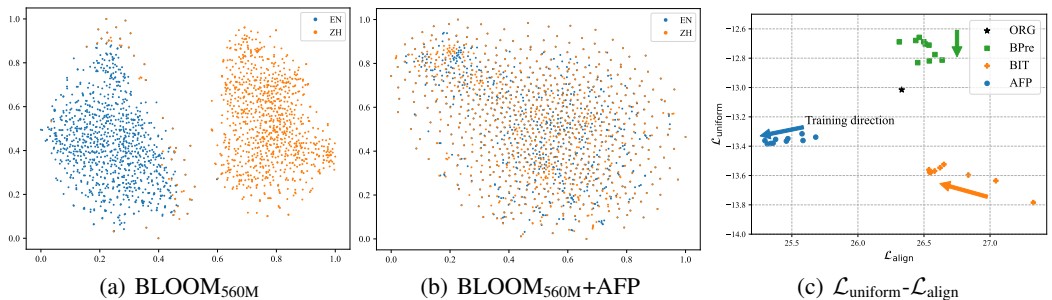

(a) BLOOM$_{560M}$       (b) BLOOM$_{560M}$+AFP       (c) $\mathcal{L}_{\text{uniform}}$-$\mathcal{L}_{\text{align}}$

Figure 3: (a, b) The t-SNE visualization of the original and aligned internal EN-ZH sentence representations of BLOOM$_{560M}$; (c) The deviation of $\mathcal{L}_{\text{uniform}}$-$\mathcal{L}_{\text{align}}$ for XGLM$_{564M}$ during training process with different multilingual training methods. The smaller these two metrics are, the better representations models learn. "BPre" and "BIT" denote bilingual pretraining and bilingual instruction tuning, respectively.

### 3.2.2 MULTILINGUAL CONTRASTIVE LEARNING ON BOTTOM LAYER PERFORMS BETTER

Figure 4(a) presents the impact of different layers applied by contrastive learning on the 5 cross-lingual datasets (XNLI, PAWS-X, XCOPA, XStoryCloze, and XWinograd). The average performance of models shows a trend of decreasing first and then increasing, which changes at the 10th layer for XGLM$_{564M}$ or the 17th layer for BLOOM$_{560M}$. And the first transformer layer is better for both models when using multilingual contrastive learning. As a result, multilingual contrastive learning is applied to the first layer after the embedding layer by default.

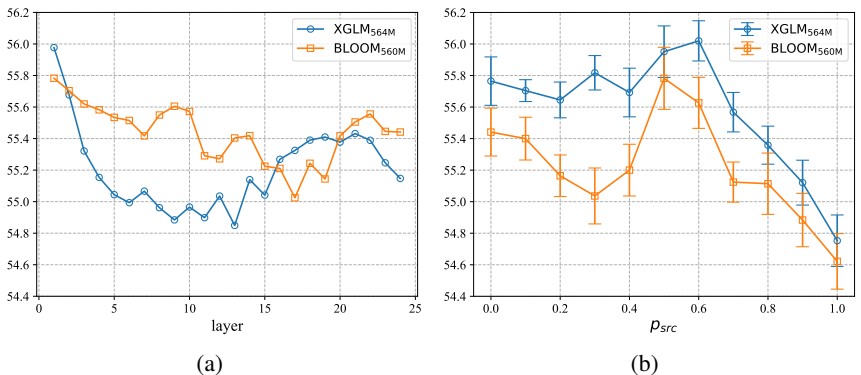

(a)            (b)

Figure 4: Effects of the target layer of MCL (a) and the $p_{src}$ of CIT (b) on 5 EN-ZH datasets.

### 3.2.3 CROSS-LINGUAL INSTRUCTION TUNING OR MONOLINGUAL INSTRUCTION TUNING?

As shown in Figure 4(b), monolingual instruction tuning for each language ($p_{src} = 1$) is inferior to cross-lingual instruction tuning ($p_{src} < 1$) for the models evaluated. Moreover, the result becomes

Table 3: In-context learning performance on NLI and Reasoning datasets across 5 languages. "**High**", "**Medium**" and "**Low**" denotes the available amount of linguistic resources. $^\dagger$ denotes the unseen language in the pre-training corpus of BLOOM. Following Lin et al. (2022), the prompt template is written in English for all languages evaluated.

| Model | XNLI | | | | | XCOPA | | | | | |
| | High | | Medium | | Low | High | | Medium | | Low | |
| | EN-0/5 | ZH-0/5 | TH$^\dagger$-0/5 | TR$^\dagger$-0/5 | SW-0/5 | EN-0/5 | ZH-0/5 | TH$^\dagger$-0/5 | TR$^\dagger$-0/5 | SW-0/5 | Avg |
|---|---|---|---|---|---|---|---|---|---|---|---|
| GPT-3$_{6.7B}$ | 55.3/52.8 | 42.4/45.9 | 38.5/36.6 | 40.5/38.4 | 34.8/33.9 | 73.6/74.5 | 55.0/57.7 | 53.7/54.4 | 53.4/53.0 | 52.3/52.1 | 49.9 |
| XGLM$_{564M}$ | 45.5/41.2 | 37.6/35.6 | 40.8/35.0 | 40.2/34.9 | 37.5/34.7 | 56.4/59.6 | 52.8/52.2 | 55.4/54.2 | 52.8/51.8 | 51.8/51.6 | 46.1 |
| +AFP | **48.0/46.3** | **42.8/42.7** | **42.8/43.3** | **40.4/42.9** | **38.9/40.0** | **60.6/61.4** | **59.0/59.4** | **59.0/60.0** | **56.6/56.0** | **57.6/55.6** | **50.7** |
| XGLM$_{1.7B}$ | 47.6/42.9 | 39.0/39.3 | 40.9/37.6 | 40.0/38.1 | 37.8/37.6 | 62.4/64.0 | 56.4/57.8 | 58.4/59.4 | 53.6/52.8 | 54.4/52.8 | 48.6 |
| +AFP | **48.5/47.7** | **43.6/44.2** | **43.2/44.0** | **42.5/41.8** | **41.0/40.4** | **65.4/66.2** | **60.8/61.6** | **62.2/60.4** | **57.6/57.0** | **59.2/59.6** | **52.3** |
| XGLM$_{7.5B}$ | 54.1/49.9 | 45.4/44.2 | 45.2/43.6 | 44.7/39.5 | 44.3/39.6 | 69.4/74.6 | 62.4/63.2 | 62.0/62.4 | 56.6/58.4 | 58.2/57.2 | 53.7 |
| +AFP | **55.8/54.1** | **50.6/48.8** | **48.1/47.2** | **46.7/44.1** | **46.1/44.2** | **71.4/75.0** | **66.8/66.6** | **63.2/64.4** | **61.8/62.0** | **62.2/62.8** | **57.1** |
| BLOOMZ$_{560M}$ | 43.8/44.5 | 41.5/40.7 | 37.8/39.2 | 35.6/35.9 | 35.8/35.8 | 54.8/57.2 | 52.0/52.8 | 52.6/52.5 | 52.6/51.8 | 52.0/52.4 | 46.1 |
| BLOOM$_{560M}$ | 44.4/40.4 | 41.1/40.3 | 33.4/35.1 | 34.5/34.1 | 35.7/34.5 | 53.0/57.4 | 49.8/54.0 | 50.8/51.8 | 52.8/52.6 | 51.2/52.0 | 44.9 |
| +AFP | **48.4/46.5** | **47.4/44.1** | **39.8/40.5** | **39.7/39.4** | **40.1/40.8** | **56.0/58.4** | **52.4/54.4** | **53.8/53.4** | **54.6/54.8** | **52.2/53.4** | **48.5** |
| BLOOMZ$_{1.7B}$ | 50.3/51.2 | 48.0/46.2 | 38.4/36.8 | 37.1/37.4 | 38.3/38.7 | 58.0/58.0 | 55.2/56.8 | 52.4/53.8 | 52.2/54.6 | 50.8/50.2 | 48.2 |
| BLOOM$_{1.7B}$ | 50.4/44.4 | 47.6/46.1 | 37.9/35.7 | 36.9/35.0 | 36.3/36.7 | 55.8/58.2 | 52.4/54.6 | 51.2/52.0 | 53.4/54.2 | 52.2/53.6 | 47.2 |
| +AFP | **52.1/51.3** | **49.1/47.1** | **41.2/41.8** | **40.1/41.3** | **41.1/42.5** | **60.2/60.4** | **55.4/58.8** | **54.2/54.6** | **55.6/56.0** | **53.6/55.0** | **50.6** |
| BLOOMZ$_{7.1B}$ | 51.1/52.0 | 49.7/48.0 | 40.9/37.6 | 39.8/36.1 | 39.2/39.7 | 61.2/62.4 | 57.6/59.8 | 53.2/51.6 | 55.0/54.2 | 53.6/52.2 | 49.7 |
| BLOOM$_{7.1B}$ | 54.0/48.7 | 48.1/47.5 | 39.5/37.4 | 38.2/35.0 | 37.7/38.9 | 58.0/58.8 | 54.0/54.8 | 52.6/52.8 | 53.8/53.4 | 53.2/54.6 | 48.6 |
| +AFP | **55.7/52.5** | **50.1/50.2** | **43.7/43.2** | **43.0/43.4** | **42.2/43.1** | **62.6/62.8** | **58.2/60.4** | **55.6/55.2** | **56.4/56.6** | **55.0/55.8** | **52.3** |

suboptimal when all samples are transferred into the cross-lingual format ($p_{src} = 0$). We empirically set the $p_{src}$ to 0.5 in the cross-lingual instruction tuning task.

## 3.3 MULTILINGUAL RESULTS

In addition to the bilingual alignment, AFP can be applied to align the models in multilingual conditions. English is first chosen as the bridge of alignment for the dominance performance in multilingual generative models. That is, the input parallel samples of AFP are selected from the EN-XX corpus, e.g., EN-ZH and EN-TH, to pull the representations and outputs of models in other languages closer to the ones in English. We also investigate the other alignment methods like pair-wise alignment in Section 3.3.1, which shows an inferior performance.

Table 3 reports the results of alignment between 5 languages from different language families (Details are reported in Appendix D), where the performance of models on the NLI and reasoning tasks is consistently improved from high-resource languages to the less-represented language Swahili. Moreover, models with AFP obtain a more balanced performance distribution. Taking XGLM models as an example, the variance of performance across 5 languages decreases from 3.32% to 2.83% on average. It is noted that AFP advances the performance of BLOOM in the two unseen languages, Thai (TH, +3.9%) and Turkish (TR, +3.92%).

Multilingual generative models also obtain a performance gain (+0.75 BLUE) in the multilingual machine translation task after alignment (Table 4). It can also find a more balanced performance distribution across languages, where the average variance reduction is 0.4 for the models evaluated.

### 3.3.1 ENGLISH AS A BRIDGE OR PAIRWISE ALIGNMENT?

Besides adopting English as a bridge to align multilingual representations, we also investigate the pairwise alignment policy, which is aligned by languages in pairs. For example, assuming to align the representations of English (EN), Chinese (ZH), and Thai (TH), the former policy comes to two parallel samples for input, which are EN-ZH and EN-TH, while the latter has three parallel samples: EN-ZH, EN-TH, and ZH-TH.

Table 4: Few-shot multilingual machine translation results on FLORES-101 devtest set. The variance of performance across the input or output languages is marked in the subscript.

| | Model | EN | ZH | TH | TR | SW | Avg |
|---|---|---|---|---|---|---|---|
| **Avg translate from the language** | XGLM$_{564M}$ | $4.8_{\pm0.8}$ | $2.1_{\pm1.7}$ | $2.2_{\pm1.8}$ | $2.2_{\pm1.8}$ | $1.1_{\pm0.9}$ | $2.5_{\pm1.7}$ |
| | +AFP | $\mathbf{5.2}_{\pm0.4}$ | $\mathbf{2.7}_{\pm1.5}$ | $\mathbf{2.8}_{\pm1.7}$ | $\mathbf{2.7}_{\pm1.6}$ | $\mathbf{2.6}_{\pm0.8}$ | $\mathbf{3.2}_{\pm1.4}$ |
| | XGLM$_{7.5B}$ | $16.3_{\pm2.2}$ | $10.0_{\pm6.5}$ | $11.0_{\pm7.0}$ | $10.0_{\pm7.3}$ | $14.0_{\pm9.8}$ | $12.3_{\pm7.5}$ |
| | +AFP | $\mathbf{16.9}_{\pm1.7}$ | $\mathbf{11.1}_{\pm6.0}$ | $\mathbf{11.6}_{\pm6.8}$ | $\mathbf{11.1}_{\pm6.9}$ | $\mathbf{14.7}_{\pm9.2}$ | $\mathbf{13.1}_{\pm7.0}$ |
| **Avg translate to the language** | XGLM$_{564M}$ | $7.4_{\pm0.7}$ | $1.3_{\pm1.0}$ | $1.4_{\pm1.1}$ | $1.4_{\pm0.9}$ | $0.9_{\pm0.7}$ | $2.5_{\pm1.7}$ |
| | +AFP | $\mathbf{8.4}_{\pm0.3}$ | $\mathbf{1.6}_{\pm0.8}$ | $\mathbf{1.8}_{\pm1.1}$ | $\mathbf{2.3}_{\pm0.8}$ | $\mathbf{1.9}_{\pm0.6}$ | $\mathbf{3.2}_{\pm1.4}$ |
| | XGLM$_{7.5B}$ | $24.3_{\pm3.3}$ | $8.0_{\pm2.9}$ | $11.0_{\pm4.3}$ | $9.6_{\pm3.7}$ | $8.5_{\pm5.7}$ | $12.3_{\pm7.5}$ |
| | +AFP | $\mathbf{24.4}_{\pm3.0}$ | $\mathbf{10.0}_{\pm2.2}$ | $\mathbf{11.8}_{\pm3.9}$ | $\mathbf{10.3}_{\pm3.2}$ | $\mathbf{9.0}_{\pm5.4}$ | $\mathbf{13.1}_{\pm7.0}$ |

Table 5: Results of different alignment methods.

| Model | EN | ZH | TH | TR | SW | Avg |
|---|---|---|---|---|---|---|
| XGLM$_{564M}$ | 50.7 | 44.6 | 46.4 | 44.9 | 43.9 | 46.1 |
| w/ EN as a bridge | **54.1** | **51.0** | **51.3** | 49.2 | 48.0 | **50.7** |
| w/ Pairwise alignment | 52.7 | 50.5 | 50.4 | **49.5** | **48.4** | 50.3 |
| BLOOM$_{560M}$ | 48.8 | 46.3 | 42.8 | 43.5 | 43.4 | 45.0 |
| w/ EN as a bridge | **52.3** | **49.6** | **46.9** | **47.1** | 46.6 | **48.5** |
| w/ Pairwise alignment | 51.6 | 48.9 | 46.5 | 46.3 | **46.8** | 48.0 |

Table 6: Prompt with semantic aligned demos.

| Model | EN | ZH | TH | TR | SW | Avg |
|---|---|---|---|---|---|---|
| XGLM$_{7.5B}$ + AFP | 64.1 | 58.2 | 55.7 | 53.7 | 53.8 | 57.1 |
| w/ Semantic aligned demos | **64.4** | **58.7** | **55.8** | **55.8** | **54.0** | **57.7** |
| BLOOM$_{7.1B}$ + AFP | 58.4 | 54.7 | 49.4 | 49.9 | 49.0 | 52.3 |
| w/ Semantic aligned demos | **58.9** | **54.8** | **49.5** | **50.2** | **49.2** | **52.5** |

The results of five languages alignment experiments on XNLI and XCOPA are reported in Table 5. The pairwise alignment policy performs consistently better in the low-resource language Swahili, although its average improvement is inferior to that when adopting English as a bridge.

### 3.3.2 COMBINATION WITH OTHER CROSS-LINGUAL METHODS

After alignment, multilingual generative models can use other cross-lingual methods for further improvement. We take a method named semantic alignment for an example, which is able to promote the cross-lingual ability using semantic aligned demos in prompt (Tanwar et al., 2023). As shown in Table 6, models obtain a further 0.4% improvement in the multilingual NLI and reasoning tasks on average.

### 3.4 EXTENDED TO ALIGNMENT IN 52 LANGUAGES

Based on the above analyses, we extend the alignment to all 52 languages in the Bactrain-X dataset by adopting English as a bridge (information about all languages used is reported in Appendix D). As shown in Table 7, models obtain a 2.6% improvement in 5 multilingual tasks on average, and mitigate the variance across languages. It is also noted that the performance of BLOOM$_{7.1B}$ on unseen languages among 5 datasets is improved by 2.8% using only parallel samples via our alignment framework, which may come from the knowledge transferred from other languages after alignment.

Table 7: In-context learning results of models on 5 datasets across all languages. The variance of performance across languages is marked in the subscript. All results are reported in Appendix B.4.

| | XNLI | | PAWS-X | | XCOPA | | XStoryCloze | | XWinograd | | |
|---|---|---|---|---|---|---|---|---|---|---|---|
| Model | 0-shot | 5-shot | 0-shot | 5-shot | 0-shot | 5-shot | 0-shot | 5-shot | 0-shot | 5-shot | Avg |
| XGLM$_{7.5B}$ | $45.6_{\pm3.4}$ | $43.6_{\pm3.1}$ | $54.7_{\pm3.1}$ | $55.1_{\pm1.6}$ | $58.9_{\pm5.0}$ | $60.4_{\pm5.7}$ | $60.6_{\pm3.9}$ | $60.5_{\pm5.0}$ | $63.9_{\pm5.1}$ | $64.7_{\pm4.2}$ | $55.3_{\pm8.5}$ |
| +AFP | $\mathbf{47.5}_{\pm3.3}$ | $\mathbf{47.7}_{\pm3.0}$ | $\mathbf{57.7}_{\pm2.3}$ | $\mathbf{57.5}_{\pm1.4}$ | $\mathbf{61.3}_{\pm4.5}$ | $\mathbf{62.4}_{\pm5.7}$ | $\mathbf{62.4}_{\pm3.7}$ | $\mathbf{63.5}_{\pm4.9}$ | $\mathbf{65.5}_{\pm4.8}$ | $\mathbf{66.7}_{\pm4.1}$ | $\mathbf{57.8}_{\pm8.0}$ |
| BLOOMZ$_{7.1B}$ | $44.1_{\pm4.0}$ | $43.5_{\pm4.6}$ | $57.8_{\pm2.6}$ | $\mathbf{56.6}_{\pm2.9}$ | $53.1_{\pm5.3}$ | $54.6_{\pm5.5}$ | $58.9_{\pm6.7}$ | $61.0_{\pm7.4}$ | $60.0_{\pm4.9}$ | $60.4_{\pm5.9}$ | $54.2_{\pm8.3}$ |
| BLOOM$_{7.1B}$ | $43.3_{\pm5.5}$ | $42.5_{\pm4.7}$ | $54.5_{\pm3.1}$ | $53.5_{\pm3.6}$ | $52.3_{\pm4.7}$ | $53.3_{\pm4.0}$ | $57.3_{\pm6.2}$ | $59.2_{\pm7.2}$ | $59.0_{\pm6.2}$ | $59.2_{\pm5.2}$ | $52.0_{\pm8.2}$ |
| +AFP | $\mathbf{45.4}_{\pm4.5}$ | $\mathbf{45.9}_{\pm3.9}$ | $\mathbf{58.1}_{\pm2.6}$ | $56.1_{\pm3.1}$ | $\mathbf{55.0}_{\pm3.7}$ | $\mathbf{55.1}_{\pm3.9}$ | $\mathbf{61.3}_{\pm6.0}$ | $\mathbf{62.5}_{\pm7.2}$ | $\mathbf{61.1}_{\pm5.8}$ | $\mathbf{60.5}_{\pm5.2}$ | $\mathbf{54.7}_{\pm8.0}$ |

## 3.5 ABLATION STUDY

To take a deep look into the improvements contributed by AFP, we conduct an ablation study on the 5 datasets of bilingual tasks using XGLM$_{564M}$ and BLOOM$_{560M}$ (Table 8).

The in-context learning abilities of the models decrease when only multilingual contrastive learning (MCL) is used. It may arise from the next word prediction ability of the model at the top layer is affected by the MCL applied to the bottom layer. Using the same data, both casual language modeling (CLM, +1.1%) and cross-lingual instruction tuning (CIT, +2.0%) can improve multilingual generative models, while the latter can promote it more. In addition, the performance of the models can be further improved after combining MCL and CIT, which is the proposed alignment framework AFP.

Table 8: Ablation study of different training methods on 5 datasets for XGLM$_{564M}$ and BLOOM$_{560M}$.

| Model | 0-shot | 3-shot | 5-shot | Model | 0-shot | 3-shot | 5-shot |
|---|---|---|---|---|---|---|---|
| XGLM$_{564M}$ | $52.94_{\pm0.54}$ | $51.71_{\pm0.90}$ | $52.03_{\pm0.89}$ | BLOOM$_{560M}$ | $51.91_{\pm0.43}$ | $52.63_{\pm0.51}$ | $52.57_{\pm0.53}$ |
| w/ MCL | $50.23_{\pm0.43}$ | $48.66_{\pm0.51}$ | $48.60_{\pm0.49}$ | w/ MCL | $52.68_{\pm0.34}$ | $52.01_{\pm0.43}$ | $51.67_{\pm0.55}$ |
| w/ CLM | $54.51_{\pm0.57}$ | $53.27_{\pm0.77}$ | $52.60_{\pm0.52}$ | w/ CLM | $53.22_{\pm0.52}$ | $53.35_{\pm0.52}$ | $53.43_{\pm0.57}$ |
| w/ CIT | $55.31_{\pm0.55}$ | $54.02_{\pm0.63}$ | $53.58_{\pm0.64}$ | w/ CIT | $54.71_{\pm0.43}$ | $54.02_{\pm0.47}$ | $54.13_{\pm0.46}$ |
| w/ AFP | $\mathbf{55.97}_{\pm0.48}$ | $\mathbf{55.50}_{\pm0.55}$ | $\mathbf{56.15}_{\pm0.43}$ | w/ AFP | $\mathbf{55.94}_{\pm0.21}$ | $\mathbf{55.04}_{\pm0.32}$ | $\mathbf{55.70}_{\pm0.31}$ |

## 4 RELATED WORK

### 4.1 MULTILINGUAL GENERATIVE LANGUAGE MODEL

Through unsupervised pre-training on the large-scale multilingual corpus, generative language models obtain impressive multilingual abilities, e.g., multilingual machine translation (Liu et al., 2020), cross-lingual natural language understanding (Xue et al., 2021) and cross-lingual in-context learning (Lin et al., 2022; Scao et al., 2022; Anil et al., 2023). Most of them extended the pre-training method developed for the monolingual corpus (Lewis et al., 2020; Raffel et al., 2020) and relied on a balanced sampling method across languages, while a significant performance gap between high-resource languages and low-represented languages persists in the pre-trained model (Asai et al., 2023). Different from the unsupervised pre-training on the multilingual corpus, this work attempts to alleviate the performance gap across languages by cross-lingual alignment using parallel samples.

### 4.2 CONTRASTIVE LEARNING IN NATURAL LANGAUGE PROCESSING

Most of the work in NLP adopted contrastive learning to improve the representation of sentences in the language model, including SentenceBERT (Reimers & Gurevych, 2019) and SimCSE (Gao et al., 2021). Specifically, contrastive learning is often applied to the representation of encoder for sentences (Pan et al., 2021), while it is less explored how to promote the representation of decoder models. In this work, we try to improve the internal multilingual representation of the Transformer decoder by multilingual contrastive learning rather than the encoder of Transformers (Vaswani et al., 2017).

## 5 CONCLUSION AND FUTURE WORK

In this paper, we proposed a simple yet effective multilingual alignment framework, including internal multilingual representations alignment and cross-lingual outputs alignment methods. Experimental results show that this framework improves both the internal representations and cross-lingual capabilities of generative models across various scales.

Beyond aligning different languages, our framework can be extended to align the internal representations and outputs across different modalities in the multi-modal generative models by replacing parallel samples. However, it is noted that the current framework relies on labeled training data for alignment. Future works can focus on the unsupervised multilingual alignment method for language models.

## 6 REPRODUCIBILITY STATEMENT

To ensure reproducibility, we report the hyper-parameters in Appendix A and the prompt templates used in all experiments in Appendix C. Considering the randomness of in-context learning, especially the few-shot one, all experiments are repeated under three random seeds, and the average performance is reported. Codes and weights will be made public after review to advocate future research.

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

## A  HYPERPARAMETERS

To align the representations and outputs of multilingual generative models, we adopt AdamW (Loshchilov & Hutter, 2019) optimizer, where $\beta_1 = 0.9$ and $\beta_2 = 0.999$, and a learning rate of 1e-5. The temperature $\tau$ is set to 0.05 in the multilingual contrastive learning task. Mixed precision training and ZeRO are applied to speed up the training process and save memory used (Micikevicius et al., 2018; Rasley et al., 2020). The number of training steps is empirically set to 10k with a batch size of 128. All experiments are conducted on a GPU server with 8*A100 80GB RAM.

## B  ADDITIONAL RESULTS

### B.1  POOLING METHODS

Given representations for each token in the sentence, there are three general methods, the last token representation, max pooling and mean pooling, to obtain the representation of this sentence. Figure 5(a) illustrates the results of XGLM$_{564M}$ under different pooling methods using AFP. It can be found that the last token and mean pooling perform better, and our method is less sensitive to the pooling method chosen. Thus, these two methods are used in AFP and are selected according to the performance of the development set.

### B.2  WEIGHT OF CROSS-LINGUAL INSTRUCTION TUNING

We find that the weight $\alpha$ of cross-lingual instruction tuning in Eq. (4) affects the multilingual performance of models. The average performance of XGLM$_{564M}$ on 5 datasets with different $\alpha$ is presented in Figure 5(b), where models perform better than the other values evaluated when $\alpha$ is set to 1.5. Therefore, we only consider a limited hyperparameter sweep for each multilingual generative model with $\alpha \in \{1, 1.5, 2\}$.

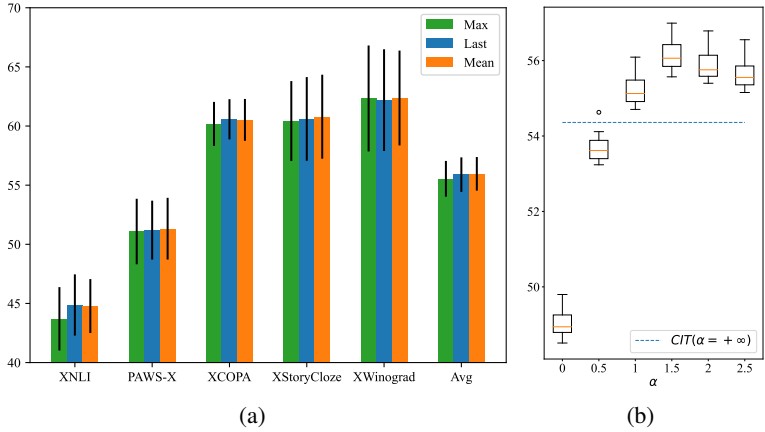

Figure 5: Results of different pooling methods (a) and weights of CIT (b) on 5 EN-ZH datasets using XGLM$_{564M}$.

### B.3 DISTRIBUTION OF MULTILINGUAL REPRESENTATIONS

Figure 6 illustrates the distributions of 5 languages sentence representations from the vanilla XGLM models and the aligned ones via t-SNE. We can find that there is a distinct gap between the sentence representations from different languages in the vanilla models (Figure 6(a)-6(c)). After alignment, the multilingual sentence distributions of models are better aligned between languages across different scales (Figure 6(d)-6(f)). The alignment and uniformity across languages in XGLM$_{7.5B}$ is not as good as the first two models, which may arise from the limited parallel samples used.

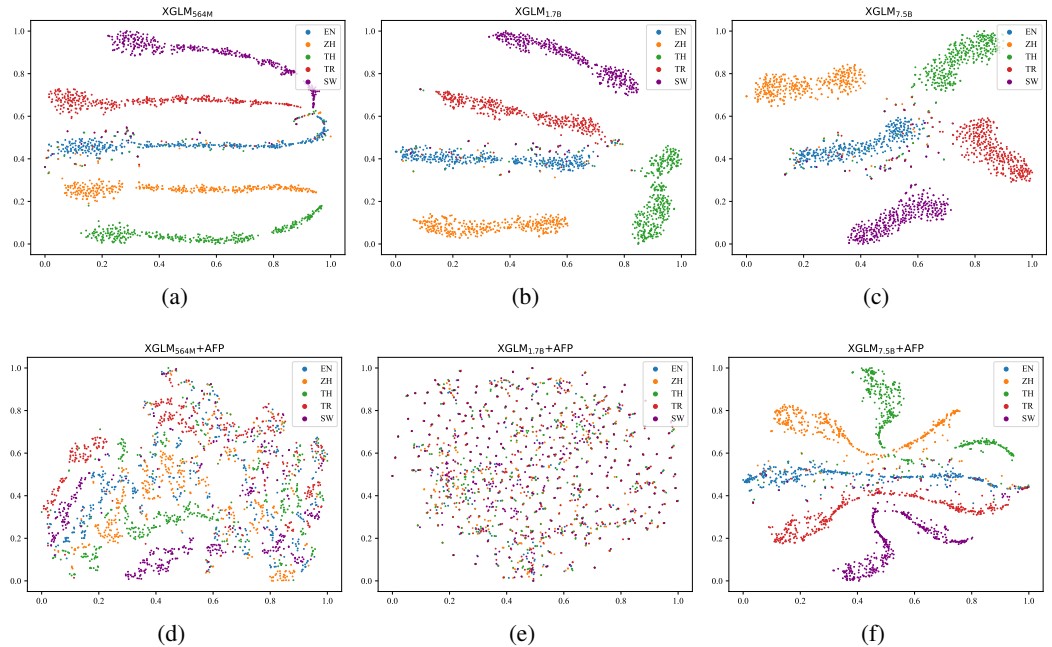

Figure 6: Distribution of multilingual sentence representations in XGLM.(Vanilla:(a)-(c), Aligned:(d)-(f), shown in t-SNE)

### B.4 PERFORMANCE ON MULTILINGUAL DATASETS

All results of XGLM$_{7.5B}$, BLOOMZ$_{7.1B}$, and BLOOM$_{7.1B}$ on the 5 multilingual datasets are reported in Tabel 9-13.

Table 9: In-context learning results on XNLI across all languages. "**High**", "**Medium**" and "**Low**" denotes the available amount of linguistic resources. [†] denotes the unseen language in the pre-training corpus of BLOOM.

| Model | #shot | High | | | | | | Medium | | | | | | Low | | | Avg |
|---|---|---|---|---|---|---|---|---|---|---|---|---|---|---|---|---|---|
| | | EN | DE[†] | ES | FR | RU[†] | ZH | AR | BG[†] | EL[†] | TH[†] | TR[†] | VI | HI | SW | UR | |
| XGLM$_{7.5B}$ | 0 | 54.1 | 42.5 | 39.9 | 49.9 | 45.0 | 45.4 | 46.4 | 48.9 | 45.4 | 45.2 | 44.7 | 47.2 | 43.2 | 44.3 | 42.1 | 45.6 |
| | 5 | 49.9 | 43.1 | 48.5 | 45.8 | 42.5 | 44.2 | 41.9 | 43.8 | 45.6 | 43.6 | 39.5 | 46.1 | 42.2 | 39.6 | 38.2 | 43.6 |
| XGLM$_{7.5B}$ +AFP | 0 | 54.8 | 44.6 | 41.9 | 51.3 | 48.4 | 50.6 | 48.8 | 48.8 | 47.2 | 48.6 | 47.4 | 47.7 | 44.4 | 45.3 | 42.6 | **47.5** |
| | 5 | 51.7 | 48.2 | 50.7 | 51.0 | 47.1 | 48.9 | 47.5 | 47.3 | 48.8 | 49.3 | 45.5 | 50.2 | 44.9 | 43.7 | 40.2 | **47.7** |
| BLOOMZ$_{7.1B}$ | 0 | 51.1 | 43.7 | 41.4 | 48.0 | 42.3 | 49.7 | 48.2 | 40.3 | 39.7 | 40.9 | 39.8 | 48.7 | 45.6 | 39.2 | 42.9 | 44.1 |
| | 5 | 52.0 | 45.3 | 43.2 | 50.2 | 41.6 | 48.0 | 45.9 | 41.3 | 38.0 | 37.6 | 36.1 | 47.7 | 45.4 | 39.7 | 40.1 | 43.5 |
| BLOOM$_{7.1B}$ | 0 | 54.0 | 39.2 | 41.5 | 51.7 | 41.3 | 48.1 | 47.4 | 37.8 | 36.3 | 39.3 | 38.9 | 48.9 | 47.4 | 37.7 | 39.9 | 43.3 |
| | 5 | 48.7 | 43.5 | 42.8 | 50.3 | 39.1 | 47.5 | 45.7 | 40.7 | 35.1 | 37.4 | 35.0 | 48.0 | 44.1 | 38.9 | 40.4 | 42.5 |
| BLOOM$_{7.1B}$ +AFP | 0 | 55.0 | 42.2 | 43.8 | 52.7 | 42.4 | 48.1 | 50.0 | 40.3 | 40.4 | 42.3 | 41.6 | 50.0 | 45.3 | 42.0 | 45.1 | **45.4** |
| | 5 | 53.3 | 44.5 | 44.1 | 51.9 | 44.1 | 49.8 | 49.2 | 42.4 | 42.3 | 41.9 | 41.3 | 50.7 | 47.0 | 42.4 | 44.1 | **45.9** |

Table 10: In-context learning results on PAWS-X across all languages. "**High**" and "**Medium**" denotes the available amount of linguistic resources. [†] denotes the unseen language in the pre-training corpus of BLOOM.

| Model | #shot | High | | | | | | Medium | Avg |
|---|---|---|---|---|---|---|---|---|---|
| | | EN | DE[†] | ES | FR | ZH | JA[†] | KO[†] | |
| XGLM$_{7.5B}$ | 0 | 58.9 | 58.0 | 57.3 | 54.0 | 52.9 | 50.3 | 51.6 | 54.7 |
| | 5 | 56.3 | 56.1 | 56.6 | 55.8 | 55.8 | 53.3 | 52.2 | 55.1 |
| XGLM$_{7.5B}$ +AFP | 0 | 61.4 | 58.2 | 58.2 | 59.4 | 57.5 | 56.2 | 53.4 | **57.7** |
| | 5 | 59.4 | 59.0 | 57.4 | 58.1 | 57.6 | 56.3 | 54.9 | **57.5** |
| BLOOMZ$_{7.1B}$ | 0 | 63.6 | 57.9 | 58.5 | 57.4 | 56.9 | 55.7 | 54.8 | 57.8 |
| | 5 | 62.2 | 56.9 | 57.3 | 57.1 | 56.1 | 54.7 | 51.8 | **56.6** |
| BLOOM$_{7.1B}$ | 0 | 59.9 | 54.7 | 57.7 | 54.0 | 53.2 | 52.0 | 50.3 | 54.5 |
| | 5 | 60.4 | 54.4 | 54.9 | 54.9 | 51.4 | 50.4 | 48.5 | 53.5 |
| BLOOM$_{7.1B}$ +AFP | 0 | 62.4 | 59.3 | 59.6 | 59.2 | 56.2 | 56.1 | 54.2 | **58.1** |
| | 5 | 62.3 | 56.2 | 55.8 | 58.0 | 53.2 | 54.9 | 52.5 | 56.1 |

Table 11: In-context learning results on XCOPA across all languages. "**High**", "**Medium**", "**Low**" and "**Ex-Low**" denotes the available amount of linguistic resources. [†] denotes the unseen language in the pre-training corpus of BLOOM.

| Model | #shot | High | | Medium | | | | | Low | | | Ex-Low | | Avg |
|---|---|---|---|---|---|---|---|---|---|---|---|---|---|---|
| | | EN | ZH | ID | IT[†] | TH[†] | TR[†] | VI | ET[†] | SW | TA | HT[†] | QU[†] | |
| XGLM$_{7.5B}$ | 0 | 69.4 | 62.4 | 63.0 | 56.0 | 62.0 | 56.6 | 61.4 | 57.4 | 58.2 | 56.2 | 56.6 | 48.0 | 58.9 |
| | 5 | 74.6 | 63.2 | 62.6 | 57.6 | 62.4 | 58.4 | 66.2 | 58.6 | 57.2 | 57.2 | 54.8 | 51.8 | 60.4 |
| XGLM$_{7.5B}$ +AFP | 0 | 71.0 | 65.2 | 64.2 | 58.6 | 64.2 | 60.2 | 63.8 | 59.8 | 60.4 | 58.0 | 58.4 | 52.2 | **61.3** |
| | 5 | 76.6 | 66.2 | 63.4 | 59.6 | 64.2 | 61.0 | 67.4 | 61.2 | 60.6 | 58.4 | 56.2 | 53.4 | **62.4** |
| BLOOMZ$_{7.1B}$ | 0 | 61.2 | 57.6 | 59.4 | 49.4 | 53.2 | 55.0 | 58.2 | 49.2 | 53.6 | 46.0 | 43.4 | 51.2 | 53.1 |
| | 5 | 62.4 | 59.8 | 61.0 | 49.4 | 51.6 | 54.2 | 61.8 | 47.2 | 52.2 | 58.4 | 48.4 | 49.2 | 54.6 |
| BLOOM$_{7.1B}$ | 0 | 58.0 | 54.2 | 59.2 | 48.6 | 52.6 | 53.8 | 59.0 | 48.0 | 53.2 | 45.0 | 46.0 | 49.4 | 52.3 |
| | 5 | 58.4 | 54.8 | 60.0 | 50.2 | 52.8 | 53.4 | 57.8 | 47.6 | 54.6 | 53.8 | 46.6 | 50.0 | 53.3 |
| BLOOM$_{7.1B}$ +AFP | 0 | 59.4 | 55.8 | 61.6 | 53.2 | 54.4 | 55.2 | 60.6 | 51.2 | 54.2 | 54.4 | 48.6 | 51.8 | **55.0** |
| | 5 | 61.0 | 56.8 | 61.0 | 51.4 | 54.2 | 54.6 | 59.6 | 49.8 | 55.2 | 57.2 | 50.2 | 50.4 | **55.1** |

Table 12: In-context learning results on XStoryCloze across all languages. "**High**", "**Medium**", "**Low**" and "**Ex-Low**" denotes the available amount of linguistic resources. $^\dagger$ denotes the unseen language in the pre-training corpus of BLOOM.

| Model | #shot | High | | | | Medium | | Low | | | Ex-Low | | Avg |
|---|---|---|---|---|---|---|---|---|---|---|---|---|---|
| | | EN | ES | RU$^\dagger$ | ZH | AR | ID | HI | SW | TE | EU | MY$^\dagger$ | |
| XGLM$_{7.5B}$ | 0 | 69.2 | 64.0 | 63.4 | 59.5 | 56.2 | 63.0 | 59.0 | 59.2 | 60.2 | 57.4 | 55.1 | 60.6 |
| | 5 | 73.7 | 63.6 | 63.6 | 59.2 | 54.4 | 62.2 | 59.4 | 58.5 | 58.7 | 56.9 | 55.7 | 60.5 |
| XGLM$_{7.5B}$ +AFP | 0 | 70.7 | 65.9 | 65.7 | 62.5 | 58.3 | 64.1 | 60.1 | 60.5 | 61.1 | 60.0 | 57.7 | **62.4** |
| | 5 | 74.7 | 67.3 | 67.4 | 62.8 | 58.2 | 67.1 | 61.4 | 61.6 | 60.7 | 59.8 | 57.6 | **62.5** |
| BLOOMZ$_{7.1B}$ | 0 | 73.7 | 64.6 | 52.6 | 62.1 | 60.3 | 62.4 | 59.2 | 55.3 | 57.7 | 51.9 | 48.4 | 58.9 |
| | 5 | 76.9 | 65.4 | 53.1 | 63.9 | 60.9 | 67.4 | 62.8 | 57.3 | 59.2 | 56.4 | 47.5 | 61.0 |
| BLOOM$_{7.1B}$ | 0 | 70.4 | 59.4 | 53.5 | 64.3 | 59.7 | 59.6 | 58.8 | 52.1 | 54.7 | 50.7 | 47.5 | 57.3 |
| | 5 | 73.5 | 64.1 | 51.7 | 64.8 | 60.0 | 64.6 | 61.3 | 53.1 | 56.9 | 53.7 | 47.3 | 59.2 |
| BLOOM$_{7.1B}$ +AFP | 0 | 70.8 | 67.6 | 54.3 | 67.6 | 62.3 | 65.2 | 62.3 | 57.6 | 57.0 | 59.2 | 50.0 | **61.3** |
| | 5 | 75.4 | 66.4 | 54.8 | 69.0 | 64.0 | 70.5 | 63.1 | 56.9 | 58.1 | 59.4 | 49.8 | **62.5** |

Table 13: In-context learning results on XWinograd across all languages. "**High**" and "**Medium**" denotes the available amount of linguistic resources. $^\dagger$ denotes the unseen language in the pre-training corpus of BLOOM.

| Model | #shot | High | | | | | Medium | Avg |
|---|---|---|---|---|---|---|---|---|
| | | EN | FR | RU$^\dagger$ | ZH | JA$^\dagger$ | PT | |
| XGLM$_{7.5B}$ | 0 | 62.8 | 59.0 | 58.7 | 73.8 | 66.4 | 62.4 | 63.9 |
| | 5 | 66.4 | 62.7 | 60.6 | 73.2 | 62.6 | 62.7 | 64.7 |
| XGLM$_{7.5B}$ +AFP | 0 | 64.6 | 61.4 | 60.3 | 75.2 | 66.6 | 64.6 | **65.5** |
| | 5 | 70.2 | 63.9 | 63.2 | 74.2 | 63.9 | 64.6 | **66.7** |
| BLOOMZ$_{7.1B}$ | 0 | 64.1 | 59.0 | 56.5 | 66.1 | 51.6 | 62.7 | 60.0 |
| | 5 | 66.9 | 60.2 | 54.3 | 68.5 | 52.6 | 60.1 | 60.4 |
| BLOOM$_{7.1B}$ | 0 | 60.6 | 56.6 | 55.2 | 71.4 | 51.7 | 58.6 | 59.0 |
| | 5 | 63.8 | 57.8 | 55.6 | 67.7 | 51.8 | 58.6 | 59.2 |
| BLOOM$_{7.1B}$ +AFP | 0 | 62.1 | 57.8 | 58.4 | 72.2 | 53.6 | 62.4 | **61.1** |
| | 5 | 64.8 | 61.4 | 56.2 | 68.5 | 52.6 | 59.7 | **60.5** |

## C  TASK DESCRIPTIONS AND PROMPT TEMPLES

To comprehensively evaluate our models, six datasets across four tasks are adopted in this work. Table 14 shows the statistics of all datasets used. It is noted that the original COPA dataset (Roemmele et al., 2011) in English is also included in the evaluation. Most temples of prompt follow the ones in Lin et al. (2022).

Table 14:  Statistic of evaluation datasets used. [‡] denotes the number of English samples, as the number of test samples in XWinograd varies across languages.

| Task | Dataset | #Lang | Data Curation | Metric | #Train | #Dev | #Test |
|------|---------|-------|---------------|--------|--------|------|-------|
| Natural Language Inference | XNLI | 15 | Translation | Accuracy | — | 2, 490 | 5, 010 |
| Paraphrase Detection | PAWS-X | 7 | Aligned | Accuracy | — | 2, 000 | 2, 000 |
| Reasoning | XCOPA | 12 | Translation | Accuracy | 33, 810 | 100 | 500 |
| | XStoryCloze | 11 | Translation | Accuracy | 361 | — | 1, 511 |
| | XWinograd | 6 | Translation | Accuracy | — | — | 2, 325[‡] |
| Multilingual Machine Translation | FLORES-101 | 101 | Aligned | BLEU | — | 997 | 1, 012 |

**Natural Language Inference**  This task aims to determine the semantic relationship between the premise and hypothesis. Table 15 shows the temple and 3-shot example used in our evaluation for this task.

Table 15:  Template and example of 3-shot demonstrations used in the evaluation of XNLI. Connectors are indicated in *italics*. The label for each example is underlined. The red text is the prediction from the model evaluated.

| Template | Candidate Verbalizer |
|----------|----------------------|
| {Premise}*, right?* {Label}, {Hypothesis} | Entailment→Yes, Neural→Also, Contradiction→No |
| *3-shot Example in English* | |
| We ask every nation to join us.*, right?* Also, We need at least 10 countries to join us.** | |
| One of the benefits we get of course is travel.*, right?* Yes, Traveling is one perk we get.** | |
| Serious crime down, but murders increase.*, right?* Yes, There has been a rise in murders.** | |
| So I'm not really sure why.*, right?* No, I am certain as to the reason why. | |

**Paraphrase Detection**  Models need to evaluate whether the second sentence is a paraphrase of the first sentence in this task. The temple and 3-shot example adopted are reported in Table 16.

**Reasoning**  Three popular multilingual reasoning datasets are applied in this task category. Given candidate sentences or pronouns mentioned above, models have to select the best one with semantic coherence and comply with the rules of the physics world. The detailed temples and examples are presented in Table 17 (XCOPA), Table 18 (XStoryCloze) and Table 19 (XWinogrande).

**Multilingual Machine Translation**  Given sentences in the source language, models for this task have to generate the corresponding sentences in the target language. Table 20 illustrates the temple and 3-shot example used in our evaluation for FLORES-101.

## D  ADDITIONAL INFORMATION ABOUT LANGUAGE CODE

Table 21 presents more information about the language codes involved in this work.

Table 16: Examples of 3-shot demonstrations used in the evaluation of PAWS-X. Connectors are indicated in *italics*. The label for each example is underlined. The red text is the prediction from the model evaluated.

| Template | Candidate Verbalizer |
|---|---|
| {Sentence 1}*, right?* {Label}, {Sentence 2} | True→Yes, False→No |
| *3-shot Example in English* | |
| Write anywhere , run once*, right?* No, Write anywhere , once run** | |
| It was Easipower that said :*, right?*Yes, It said that Easipower was ,** | |
| In 1951 , he died and retired in 1956 .*, right?* No, He died in 1951 and retired in 1956 .** | |
| Green took over Park 's No .*, right?* Yes, Park Green took over No . | |

Table 17: Examples of 3-shot demonstrations used in the evaluation of XCOPA. Connectors are indicated in *italics*. The label for each example is underlined. The red text is the prediction from the model evaluated.

| Template | Candidate Verbalizer |
|---|---|
| [*cause:*|*effect:*] {Sentence 1} [*because*|*so*] {Label} | Identity |
| *3-shot Example in English* | |
| *cause:* The woman resigned.*because* She thinks her boss is behaving immorally.** | |
| *effect:* I pulled the rubber band.*so* It stretches out.** | |
| *cause:* My skin suddenly broke out in a rash.*because* I came across poison ivy in my yard.** | |
| *cause:* The girl pinched her nose.*because* The baby soiled the diaper. | |

Table 18:  Examples of 3-shot demonstrations used in the evaluation of XStoryCloze. Connectors are indicated in *italics*. The label for each example is underlined. The red text is the prediction from the model evaluated.

| Template | Candidate Verbalizer |
|---|---|
| {Sentence 1} {Sentence 2} {Sentence 3} {Sentence 4} {Label} | Identity |

*3-shot Example in English*

Ava started to notice wrinkles by her eyes. She bought an expensive wrinkle cream. She applied it every night. After a month she checked her eyes out carefully. She was happy to see her wrinkles were gone.**

Jenny wanted to learn how to ride a horse. She went to a local horse farm. After a quick lesson, she mounted the horse. A feeling of joy enveloped her as she rode the horse around a ring. She decided to come back soon for another fun lesson.**

Rick liked eating chocolate oatmeal. But his friend suggested that he use higher quality cocoa powder. Rick was tight about money. But he decided to buy more expensive cocoa powder just once. The taste was worth the price.**

Gordon bought his son a remote control car for Christmas. But he realized that it needed AA batteries. Gordon could not find any. So the next day, he went to the toy store where he bought the car. He bought a big package of AA batteries.

Table 19:  Examples of 3-shot demonstrations used in the evaluation of XWinogrande. Connectors are indicated in *italics*. The label for each example is underlined. The red text is the prediction from the model evaluated.

| Template | Candidate Verbalizer |
|---|---|
| {Part 1 of Sentence} {Label} {Part 2 of Sentence} | Identity |

*3-shot Example in English*

Charles Dickinson shot at Andrew Jackson, so Charles Dickinson started reloading.**

The cheetah outran the antelope so The cheetah got to eat.**

The lawyer asked the witness a question, but The lawyer was reluctant to repeat it.**

The outlet powered the lamp when The outlet had electricity.

Table 20: Examples of 3-shot demonstrations used in the evaluation of FLORES-101. Connectors are indicated in *italics*. The label for each example is underlined. The red text is the prediction from the model evaluated.

| **Template** | **Candidate Verbalizer** |
|---|---|
| {Src. Lang.}*:* {Src. Sent.} = {Tgt. Lang.}*:* {Tgt. Sent.} | Identity |

*3-shot Example in English*

English*:* Since moving to the Catalan-capital, Vidal had played 49 games for the club. = French*:* Depuis son arrivée dans la capitale catalane, Vidal a joué 49 matchs pour le club.**

English*:* Nadal's head to head record against the Canadian is 7–2. = French*:* Le score de Nadal en confrontations directes face au Canadien est de 7-2.**

English*:* He recently lost against Raonic in the Brisbane Open. = French*:* Il a récemment perdu un match contre Raonic durant l'Open de Brisbane.**

English*:* Piquet Jr. was sacked after the 2009 Hungarian Grand Prix. = French*:* Piquet Jr. a été limogé après le Grand Prix de Hongrie 2009.

Table 21: Details of Language codes in this work. $\star$ denotes the language used in bilingual and 5-language experiments. $\dagger$ indicates the languages involved in the multilingual evaluation datasets but not in Bactrian-X.

| ISO 639-1 | Language | Family | | ISO 639-1 | Language | Family |
|---|---|---|---|---|---|---|
| AF | Afrikaans | Indo-European | | LT | Lithuanian | Indo-European |
| AR | Arabic | Afro-Asiatic | | LV | Latvian | Indo-European |
| AZ | Azerbaijani | Turkic | | MK | Macedonian | Indo-European |
| BG$^\dagger$ | Bulgarian | Indo-European | | ML | Malayalam | Dravidian |
| BN | Bengali | Indo-European | | MN | Mongolian | Mongolic |
| CS | Czech | Indo-European | | MR | Marathi | Indo-European |
| DE | German | Indo-European | | MY | Burmese | Sino-Tibetan |
| EL$^\dagger$ | Greek, Modern | Indo-European | | NE | Nepali | Indo-European |
| EN$^\star$ | English | Indo-European | | NL | Dutch | Indo-European |
| ES | Spanish | Indo-European | | PL | Polish | Indo-European |
| ET | Estonian | Uralic | | PS | Pashto | Indo-European |
| EU$^\dagger$ | Basque | Language Isolate | | PT | Portuguese | Indo-European |
| FA | Persian | Indo-European | | QU$^\dagger$ | Quechua | - |
| FI | Finnish | Uralic | | RO | Romanian | Indo-European |
| FR | French | Indo-European | | RU | Russian | Indo-European |
| GL | Galician | Indo-European | | SI | Sinhala | Indo-European |
| GU | Gujarati | Indo-European | | SL | Slovene | Indo-European |
| HE | Hebrew | Afro-Asiatic | | SV | Swedish | Indo-European |
| HI | Hindi | Indo-European | | SW$^\star$ | Swahili | Niger-Congo |
| HR | Croatian | Indo-European | | TA | Tamil | Dravidian |
| HT$^\dagger$ | Haitian Creole | French Creole | | TE | Telugu | Dravidian |
| ID | Indonesian | Austronesian | | TH$^\star$ | Thai | Kra-Dai |
| IT | Italian | Indo-European | | TL | Tagalog | Austronesian |
| JA | Japanese | Japonic | | TR$^\star$ | Turkish | Turkic |
| KA | Georgian | Kartvelian | | UK | Ukrainian | Indo-European |
| KK | Kazakh | Turkic | | UR | Urdu | Indo-European |
| KM | Khmer | Austroasiatic | | VI | Vietnamese | Austroasiatic |
| KO | Korean | Koreanic | | XH | Xhosa | Niger-Congo |
| | | | | ZH$^\star$ | Chinese | Sino-Tibetan |

