# OpenReview forum: "Align after Pre-train: Improving Multilingual Generative Models with Cross-lingual Alignment"
_ICLR.cc/2024/Conference — ICLR 2024 Conference Withdrawn Submission_

### Official Review · Reviewer_ehYp · 2023-10-24

**Soundness:** 3 good
**Presentation:** 3 good
**Contribution:** 2 fair
**Rating:** 5
**Confidence:** 4

**Summary:**

This paper introduce a simple and effective multilingual alignment framework (AFP), there are two major components: 1. multilingual contrastive learning on internal representation, 2 Cross-lingual instruction tuning on the outputs.  Through the alignment, the author argues that the performance of low-resource languages will be improved through the knowledge transfer.  Experiments on 52 languages show that  with a small number of training tokens, the propose method could improve the cross-lingual ability of generative models.

**Strengths:**

The paper is clearly written and easy to understand.
The proposed MCL and CIT method is solid in improving the cross-lingual ability of pre-trained language models.
And I like the experiment design and analysis: 1. Extensive experiment on 52 languages to show the cross-lingual ability, also got good results on unseen languages Thai and Turkish . 2. use embedding distance and uniformity to measure the distribution of multilingual representation.
The content in Appendix will make it easier for others to reimplement this work.

**Weaknesses:**

1. I think the innovativeness of this paper is limited:  the proposed multilingual contrastive learning on internal representation is not new, as the author listed in Sec2.1 Para 1. Also in "On learning universal representations across languages" Wei. ICLR2020, the proposed contrastive learning method for universal representation learning is similar.
    On the other hand, the cross-lingual instruction tuning also has some similar work: like "Few-shot Learning with Multilingual Generative Language Models" use cross-lingual demonstrations in the tuning process.  As there are no comparison system in the main experiment, I suggest the author add one or two related work to better show the merits of this work.

2. A small issue with the machine translation experiment,  I think the parallel data used in AFP framework should be added into the baseline model through fine-tuning or demonstrations in prompt for a fair comparison.  And the improvement in BLEU is marginal to me.
So I am a little skeptical on the improvement of MT task.   Also a typo in Sec 3.3 BLUE --> BLEU

**Questions:**

Please refer to the weakness for context:
1. Is there any related work in cross-lingual instruct tuning that could be added as the compare system?
2. For the MT experiment, is it possible to add the parallel data used in AFP framework into the baseline model for a fair comparison?

---

> ### Author Response · Authors · 2023-11-16
> **Response to Reviewer ehYp**
>
> Thank you for your detailed review! You evidently spent a substantial time reviewing our paper, and we are thankful for your efforts during the challenging review period. We will explain your concerns point by point.
>
> > Weakness 1: The innovativeness of this paper is limited.
>
> We acknowledge that the concepts of cross-lingual contrastive learning and instruction tuning are not novel.
> The novelty of this work comes from the following two parts:
>
> (1). Natural language generation task is different from natural language understanding task, there are fewer studies to investigate effectiveness of the contrastive learning in generation tasks. In this work, we find that using multilingual contrastive learning alone **cannot** enhance the cross-lingual ability of multilingual generative model, and in most cases, it will impair the performance of the model, e.g., the inferior results in the 'w/MCL' row of Table 8 in the ablation experiment (Section 3.5). We argue that the generation ability of the model may be affected by the multilingual contrastive learning method.
>
>
> (2). Our cross-lingual instruction fine-tuning differs from multilingual instruction tuning, where models usually learn and reply in the same language. It requires models to **reply in different languages**. The work mentioned, which is "Few-shot Learning with Multilingual Generative Language Models"[1], investigates the impact of **prompt temple** in the demonstrations, such as same-language-**prompting** and source-language-prompting, rather than the instruction tuning process.
>
> > Weakness 1&Question 1: Is there any related work in cross-lingual instruct tuning that could be added as the compare system?
>
> Of course, we add the CoIT (Cross-lingual Instruction Tuning) method of xLLaMA[2] for comparison using the same training data in EN-ZH bilingual instruction tuning, and the experimental results are as follows:
>
> |   **Model**  | EN-0/5(XNLI) | ZH-0/5(XNLI) | EN-0/5(PAWS-X) | ZH-0/5(PAWS-X) |  Avg |
> |:------------:|:------------:|:------------:|:--------------:|:--------------:|:----:|
> | $Llama_{7b}$ |   54.5/49.0  |   45.9/44.9  |    67.8/64.2   |    55.4/53.1   | 54.4 |
> |     +CoIT    |   54.8/50.0  |   46.2/47.2  |    68.3/63.8   |    57.5/55.2   | 55.4 |
> |     +AFP     |   55.9/54.1  |   47.6/48.4  |    70.0/64.3   |    58.6/56.1   | 56.9 |
>
> Table 1. In-context learning performance on XNLI and PAWS-X.
>
> It can be found that AFP achieves a higher improvement than CoIT.
> The difference between AFP and CoIT lies in the additional multilingual contrastive learning and cross-lingual answering format, which is in line with findings in the ablation study that multilingual contrastive learning further boosts the performance of cross-lingual instruction tuning.
>
>
> > Weakness 2&Question 2: For the MT experiment, is it possible to add the parallel data used in AFP framework into the baseline model for a fair comparison?
>
> Sure, we supplement the results of the baseline model fine-tuned on the parallel data ('+FT'):
>
> |     Model     |  0/1/5/10(EN->ZH)  |  0/1/5/10(ZH->EN)  |      0/1/5/10(Avg)     |
> |:-------------:|:------------------:|:------------------:|:----------------------:|
> | $XGLM_{564M}$ |   0.0/1.0/4.2/4.2  |   0.0/6.4/8.1/8.3  |     0.0/3.7/6.0/6.3    |
> |      +FT      |   0.8/3.3/4.5/4.8  |   3.1/7.9/8.5/8.8  |     2.0/5.6/6.5/6.8    |
> |      +AFP     |   2.0/4.2/5.3/5.9  |   5.3/8.7/9.0/9.2  |   **3.7/6.5/7.2/7.6**  |
> | $XGLM_{7.5B}$ | 0.0/13.6/13.8/13.9 | 0.1/19.2/19.5/20.1 |   0.1/16.4/16.7/17.0   |
> |      +FT      | 1.2/13.9/14.3/14.5 | 4.4/19.3/19.6/20.2 |   2.8/16.6/17.0/17.4   |
> |      +AFP     | 2.5/14.5/14.8/15.0 | 6.7/19.5/19.6/20.3 | **4.6/17.0/17.2/17.7** |
>
> Table 2. Translation results on FLORES-101 devtest set(ZH-EN).
>
> It can be found that fine-tuning on the same parallel data benefits the machine translation task (+1.1 BLEU), but AFP can bring further improvement(+1.9 BLEU).
>
> > Typo in Sec 3.3: BLUE --> BLEU
>
> Thank you for reminding us! We will correct this typo in the next edition.
>
> [1] Lin, Xi Victoria, et al. "Few-shot Learning with Multilingual Generative Language Models." EMNLP 2022.
>
> [2] Zhu, Wenhao, et al. "Extrapolating Large Language Models to Non-English by Aligning Languages." arXiv:2308.04948 (2023).

---

> > ### Comment · Reviewer_ehYp · 2023-11-21
> >
> > I have read the author response and the reviews.  Give credit to the author for adding two comparison experiments in a short time, the results show that CoIT and +FT do brings some improvement over the baseline, the proposed AFP is better or marginally better than the compared system.
> > At the same time, I shared the view with Review 1odi that the innovativeness of this paper is limited, and the response on this point is not quite convincing to me, So I choose to lower my score to 5.

---

> > > ### Author Response · Authors · 2023-11-23
> > >
> > > Dear Reviewer ehYp,
> > >
> > > We are sorry that our response made you misunderstand the innovation of the paper. Now Reviewer 1odi has reached an agreement on the finding that "using multilingual contrastive learning alone cannot enhance the cross-lingual ability of multilingual generative model" is a good point, and updates the rating. Are there any other concerns? We will try our best to solve any concerns you have.
> > >
> > > We sincerely thank you for your feedback. Thank you again for your time and effort in reviewing this manuscript!

---

### Official Review · Reviewer_GUFo · 2023-10-30

**Soundness:** 2 fair
**Presentation:** 3 good
**Contribution:** 2 fair
**Rating:** 3
**Confidence:** 4

**Summary:**

The paper proposes align after pretrain (AFP) method to improve multilingual capabilities of LLM. AFP consists of two training objectives:  (1) contrastive learning objective to align embedding spaces and (2) cross-lingual instruction objective to improve cross-lingual generation. AFP is evaluated on 4 tasks: natural language inference, paraphrase detection, reasoning and machine translation. The evaluation using XGLM and BLOOM models show that when AFP is used, it boosts the multilingual performances of the base models.

**Strengths:**

- The paper combines two popular ideas in NLP to improve multilingual capabilities of LLM. Although the idea of making pretrained LM more multilingual is not new. [Earlier work](https://arxiv.org/abs/2002.07306)  has shown the extreme of making bilingual LM out of English only LM.
- The AFP method is simple, and it should work for languages that have parallel data.
- Analysis of embeddings after AFP

**Weaknesses:**

- The motivation for AFP is because current public LLMs are English centric, thus I wonder the value of this approach and its impact compared to training a native multilingual LLM from scratch such as [PolyLM](https://arxiv.org/abs/2307.06018) , which is multilingual by design. I think it’s important to have PolyLM in the evaluation to understand the gap in multilingual capabilities  and where AFP stands.

- Evaluated tasks are simple (except few-shot machine translation). XNLI, PAW-S, XCOPA, XStorzyCloze, and XWinograd don’t seem to test the resulting model’s ability to generate languages. For instance, for XNLI, the model only needs to generate one token (yes/also/no), for XCOPA, for reasoning tasks, the model doesn’t need to generate the model, instead it is used to score the answers and find the best one. Thus I think that the generative abilities in multilingual setup are not properly measured. I wonder if other multilingual tasks such as summarization/QA are more appropriate for evaluation.

- The crosslingual finetuning step leverages machine translation outputs, which is prone to error.

- While the evaluation is done based on the existing cross-lingual datasets, which is perhaps outdated in the age of LLMs with emergent abilities. In order to advance multilingual ability, i think the first step (and also the very important one) is to have an adequate  multilingual benchmark for LLMs. Without that, it's difficult to assess any claim about  improving multilingual generative models.
- The cross lingual finetuning step leverages machine translation outputs, which is prone to error. LLMs are known for hallucinating their generation, thus cross-lingual finetuning could make it even worse for generation in non-English languages. Does AFP cause more hallucination in language generation? Can you measure that? And what is the strategy to prevent such cases.

**Questions:**

See the question in the above section.

---

> ### Author Response · Authors · 2023-11-16
> **Response to Reviewer GUFo (1/2)**
>
> Thank you for your review! We will explain your concerns point by point.
>
> > Weakness 1: The motivation for AFP and its value.
>
> There may be a misunderstanding here. The motivation of AFP lies in the performance bias toward high-resource languages, which is also found in the native multilingual LLMs from scratch, including **XGLM, BLOOM, and the mentioned PolyLM[1]**.
> Experimental results demonstrate that our method greatly improves the cross-lingual ability of generative models, including multilingual ones (XGLM and BLOOM) and models pre-trained on English corpus (LLaMA), by using less than 1M parallel samples.
>
>
> Thank you for your recommendation to analyze PolyLM.
> And we conduct preliminary bilingual alignment experiments on $PolyLM_{1.7B}$:
>
> |      Model      | EN-0/5(XNLI) | ZH-0/5(XNLI) | EN-0/5(PAWS-X) | ZH-0/5(PAWS-X) | Avg  |
> |:---------------:|:------------:|:------------:|:--------------:|----------------|------|
> | $PolyLM_{1.7B}$ |   48.6/47.3  |   40.6/40.3  |    52.4/50.9   | 51.5/53.6      | 48.1 |
> |       +AFP      |   49.4/47.7  |   43.6/43.0  |    58.0/56.2   | 56.4/56.1      | 51.3 |
>
> Table 1. In-context learning performance on XNLI and PAWS-X.
>
> |      Model      | EN-0/5(XCOPA) | ZH-0/5(XCOPA) | EN-0/5(XStoryCloze) | ZH-0/5(XStoryCloze) | EN-0/5(XWinograd) | ZH-0/5(XStoryCloze) |  Avg |
> |:---------------:|:-------------:|:-------------:|:-------------------:|:-------------------:|:-----------------:|:-------------------:|:----:|
> | $PolyLM_{1.7B}$ |   56.6/59.6   |   55.0/55.8   |      61.0/60.4      |      54.9/54.2      |     54.0/56.7     |      63.5/63.9      | 58.0 |
> |       +AFP      |   59.6/60.4   |   58.2/58.2   |      62.0/60.8      |      54.9/54.3      |     54.5/56.9     |      64.7/66.1      | 59.2 |
>
> Table 2. In-context learning performance on XCOPA, XStoryCloze, and XWinograd.
>
> We find that AFP improves the cross-lingual performance of $PolyLM_{1.7B}$ (+2.0% on average), which further proves the effectiveness of our method.
>
> > Weakness 2: Evaluated tasks are simple. The generative abilities in multilingual setup are not properly measured. Are other multilingual tasks, such as summarization/QA, more appropriate for evaluation?
>
> We acknowledge that the five datasets you mentioned are simpler than generation tasks.
> However, current multilingual generative models have not achieved satisfactory results on these simpler datasets, where the best result in this paper is still lower than 90%.
> It means that they can still reflect the improvement in the cross-lingual performance of the model, especially in the ability to cross-lingual understanding.
> Moreover, similar datasets such as MMLU are widely adopted to evaluate LLMs.
>
>
> Although the generation task can better reflect the generation ability of language models, the evaluation of the generation task, e.g., open domain QA, is still an open question, especially in the cross-lingual condition.
> Therefore, we only adopt a commonly used generation task, which is machine translation, to evaluate the cross-lingual generation ability of the model.
> In addition, in order to better reflect the improvement in cross-lingual generation ability, we supplement the evaluation results of cross-lingual summarization tasks:
>
> |     Model     | EN->ZH | ZH->EN |  Avg |
> |:-------------:|:------:|:------:|:----:|
> | $XGLM_{7.5B}$ |  3.13  |  3.71  | 3.42 |
> |      +AFP     |  3.43  |  5.84  | 4.64 |
>
> Table 3. One-shot in-context learning performance on EN-ZH cross-lingual summarization task[2] (Rouge1).
>
> In line with the results of the machine translation task, $XGLM_{7.5B}$ also obtains an improvement on the cross-lingual summarization task after using AFP.
>
> > Weakness 3: The cross-lingual finetuning step leverages machine translation outputs, which is prone to error.
>
> Thank you for reminding us! In future work, we will add a data evaluation process to reduce the error from machine translation.
>
> > Weakness 4: The existing cross-lingual datasets are perhaps outdated in the age of LLMs with emergent abilities.
>
> Similar to the response above, current multilingual generative models have not achieved satisfactory results on these datasets, where the best result in this paper is still lower than 90%.
> It means that they can still reflect the improvement in the cross-lingual performance of the model, especially in the ability to cross-lingual understanding.
>
> Moreover, thank you for your suggestion on the research of multilingual generation model!
> We agree that it is important to build an adequate multilingual benchmark for LLMs, which is part of our future work.
>
> [1] Wei, Xiangpeng, et al. "Polylm: An open source polyglot large language model." arXiv:2307.06018 (2023).
>
> [2] Zhu, Junnan, et al. "NCLS: Neural Cross-Lingual Summarization." EMNLP-IJCNLP 2019.

---

> > ### Author Response · Authors · 2023-11-16
> > **Response to Reviewer GUFo (2/2)**
> >
> > > Weakness 5: Does AFP cause more hallucination in language generation? Can you measure that? And what is the strategy to prevent such cases.
> >
> > Thanks for reminding us of the hallucination phenomenon in language generation.
> > There are many kinds of hallucination generated from LLM, and we choose the "source language hallucination problem" [3,4] to analyze, which is more relevant and important for multilingual generative models.
> > In this kind of hallucination, model may overfit to the training languages and partially forget their generation ability in the target language, generating text in the wrong language[3].
> > Given 1k questions randomly sampled from HC3 datasets[5], we count the proportion of the responses in the target language when the model is prompted to conduct a cross-lingual response:
> >
> > |     Model     | 0/1-shot(EN->ZH) | 0/1-shot(ZH->EN) |  Avg |
> > |:-------------:|:----------------:|:----------------:|:----:|
> > | $XGLM_{7.5B}$ |     40.3/74.2    |      1.7/8.2     | 31.1 |
> > |      +AFP     |     78.8/84.4    |     64.6/76.3    | 76.0 |
> >
> > Table 4. Percentages of target languages in the generated text. The higher the percentage, the less source language hallucination.
> >
> > Table 4 shows that the percentage of target languages in the generated text increased by 44.9%, which indicates that the source language hallucination problem is mitigated by AFP.
> >
> > As for how to further alleviate the source language hallucination problem, we think it is more appropriate to use external assessment and feedback in multilingual generation model for the language detection of text is simpler than fact verification.
> >
> > [3] Vu, Tu, et al. "Overcoming Catastrophic Forgetting in Zero-Shot Cross-Lingual Generation." EMNLP 2022.
> >
> > [4] Pfeiffer, Jonas, et al. "mmT5: Modular Multilingual Pre-Training Solves Source Language Hallucinations." arXiv:2305.14224 (2023).
> >
> > [5] Guo, Biyang, et al. "How close is chatgpt to human experts? comparison corpus, evaluation, and detection." arXiv:2301.07597 (2023).

---

> ### Author Response · Authors · 2023-11-23
> **Kind Remainder**
>
> Dear Reviewer GUFo,
>
> We wish to express our sincere gratitude once again to you for the feedback. We would like to gently bring to your attention that the discussion phase between authors and reviewers is closing to the end (within 12 hours).
>
> We hope our response has addressed your concerns. Should you have any further insights to share, we are more than willing to sustain our discussion until the deadline.

---

### Official Review · Reviewer_1odi · 2023-11-02

**Soundness:** 3 good
**Presentation:** 3 good
**Contribution:** 2 fair
**Rating:** 6
**Confidence:** 4

**Summary:**

The paper proposes align after pre-train (AFT), a cross-lingual alignment framework that enhances cross-lingual abilities of multilingual generative language models. AFT utilizes translation pairs to align the internal sentence representations across languages, through multilingual contrastive learning. Besides, AFT performs cross-lingual instruction tuning that makes the language model respond in target languages given source language prompts. With two objectives combined, the authors trained multilingual LMs on the basis of several multilingual LLMs. The experimental results demonstrate that the alignment greatly improves the cross-lingual ability of the models on several multilingual tasks under zero-shot and few-shot settings.

**Strengths:**

- The paper conducts extensive experiments where AFT is evaluated on various multilingual LLMs (XGLM, BLOOM, and Llama) on four types of multilingual tasks. The results show that AFT consistently improves the LLMs across models, tasks, and setups. Besides, the AFT is also evaluated beyond the bilingual setup, with the alignment extended to 52 languages.
- The provides ablation studies on the training objectives and different alignment methods. The ablation studies support the effectiveness of multilingual contrastive learning and cross-lingual instruction tuning when they are used together.
- The paper presents visualization of the natural language representations from different languages, and demonstrates the alignment effects on the hidden representations inside LLMs.

**Weaknesses:**

- The novelty of the proposed method is limited. The AFT framework is a combination of cross-lingual contrastive learning and instruction tuning objectives, both of which have been shown to be effective for enhancing cross-lingual abilities in related works. (1) InfoXLM[1] utilizes cross-lingual contrastive learning with translation pairs to enhance multilingual LMs, and demonstrates its effectiveness in improving cross-lingual transferability. The difference is whether to put the contrastive object to MLM-trained models or  CLM-trained models. (2) BLOOMZ[2] applies multilingual multitask finetuning (a.k.a. instruction tuning) and observes better zero-shot performance.
-  In section 2.2, the paper claims that the proposed cross-lingual instruction tuning (CIT) is proposed to further align the outputs, which is more difficult than the multilingual instruction tuning. However, I did not find ablations to support this. I would guess the gain is mainly from instruction tuning instead of its cross-lingual alignment effect of CIT.
- Insufficient literature review in the related work section.

[1] InfoXLM: An Information-Theoretic Framework for Cross-Lingual Language Model Pre-Training

[2] Crosslingual Generalization through Multitask Finetuning

**Questions:**

- Does cross-lingual instruction tuning work better than multilingual instruction tuning?

After applying machine translation to the instruction tuning data, you could obtain at least twice the training data (N times training data for N target languages). It is unclear why multilingual instruction tuning, which has N times data, would perform worse than the proposed cross-lingual instruction tuning.

---

> ### Author Response · Authors · 2023-11-16
> **Response to Reviewer 1odi (1/2)**
>
> Thank you for your detailed review! You evidently spent a substantial time reviewing our paper, and we are thankful for your efforts during the challenging review period. We will explain your concerns point by point.
>
> > Weakness 1: The novelty of the proposed method is limited.
>
> We acknowledge that the concepts of cross-lingual contrastive learning and instruction tuning are not novel.
> The novelty of this work comes from the following two parts:
>
> (1). Natural language generation task is different from natural language understanding task, there are fewer studies to investigate effectiveness of the contrastive learning in generation tasks. In this work, we find that using multilingual contrastive learning alone **cannot** enhance the cross-lingual ability of multilingual generative model, and in most cases, it will impair the performance of the model, e.g., the inferior results in the 'w/MCL' row of Table 8 in the ablation experiment (Section 3.5). We argue that the generation ability of the model may be affected by the multilingual contrastive learning method.
>
> (2). Our cross-lingual instruction fine-tuning differs from multilingual instruction tuning, where models usually learn and reply in the same language. It requires models to **reply in different languages**.
>
> The experimental results prove the effectiveness of the cross-lingual instruction tuning task:
>
> (1). The cross-lingual performance of BLOOM model with 167k parallel corpus surpasses that of BLOOMZ (Table 1, 3, and 7), which is trained with 78M multilingual instructions.
>
> (2). As shown in Figure 4(b), given the EN-ZH bilingual samples, the results of fine-tuning multilingual instruction tuning ($p_{src}$=1) are worse than that of fully cross-lingual instruction tuning ($p_{src}$=0).
>
> > Weakness 2&Question 1: Does cross-lingual instruction tuning work better than multilingual instruction tuning?
>
> According to our empirical results on $BLOOM_{560M}$ and $XGLM_{564M}$, cross-lingual instruction tuning performs better than multilingual instruction tuning.
>
> Firstly, under the condition of EN-ZH bilingual instruction tuning, we compare the results of multilingual instruction tuning ($p_{src}$=1, which is also called monolingual instruction tuning for each language) and fully cross-lingual instruction tuning ($p_{src}$=0).
> We find that the performance of cross-lingual instruction fine-tuning is 0.9% higher on average (Figure 4(b) in Section 3.2.3).
>
> To further prove the effectiveness of cross-lingual tuning(CIT), we supplement the results of model tuning with 5 times multilingual instruction-tuning(MIT) samples in the multilingual condition of five languages for comparison(EN, ZH, TH, TR, SW):
>
> | Model     | EN-0/5    | ZH-0/5    | TH-0/5    | TR-0/5    | SW-0/5    | Avg |
> |---------------|-----------|-----------|-----------|-----------|-----------|-----|
> | $XGLM_{564M}$ | 45.5/41.2 | 37.6/35.6 | 40.8/35.0 | 40.2/34.9 | 37.5/34.7 |  38.3  |
> | +MIT          | 46.8/43.4 | 41.5/40.8 | 41.6/42.6 | 40.4/38.7 | 38.3/37.8 |  41.2  |
> | +CIT          | 47.8/44.9 | 42.2/41.3 | 42.4/42.8 | 40.2/40.1 | 38.6/38.9 |  **41.9**  |
> | $BLOOM_{560M}$ | 44.4/40.4 | 41.1/40.3 | 33.4/35.1 | 34.5/34.1 | 35.7/34.5 | 37.4 |
> | +MIT           | 45.5/42.6 | 44.9/42.7 | 37.8/39.3 | 38.5/38.0 | 38.7/37.9 | 40.6 |
> | +CIT           | 47.2/44.7 | 46.2/43.1 | 38.7/40.1 | 38.7/38.5 | 39.3/38.6 | **41.5** |
>
> Table 1. In-context learning performance on XNLI.
>
> | Model     | EN-0/5    | ZH-0/5    | TH-0/5    | TR-0/5    | SW-0/5    | Avg |
> |---------------|-----------|-----------|-----------|-----------|-----------|-----|
> | $XGLM_{564M}$ | 56.4/59.6 | 52.8/52.2 | 55.4/54.2 | 52.8/51.8 | 51.8/51.6 | 53.9 |
> | +MIT          | 58.2/60.2 | 57.8/56.8 | 58.8/58.0 | 55.4/55.0 | 56.0/54.2 | 57.0 |
> | +CIT          | 60.2/60.8 | 58.6/58.4 | 58.8/58.6 | 55.6/55.2 | 56.6/56.2 | **57.9** |
> | $BLOOM_{560M}$ | 53.0/57.4 | 49.8/54.0 | 50.8/51.8 | 52.8/52.6 | 51.2/52.0 | 52.5 |
> | +MIT           | 54.6/57.2 | 51.8/53.0 | 53.0/52.8 | 53.4/53.6 | 53.0/54.0 | 53.6 |
> | +CIT           | 55.8/57.8 | 52.2/54.2 | 53.6/53.2 | 54.0/53.8 | 52.0/53.2 | **54.0** |
>
> Table 2. In-context learning performance on XCOPA.
>
> From the above results, we can find that cross-lingual instruction tuning is still better than multilingual instruction tuning by 0.7% on average.
>
> As for the influence of the sample size, Chen et al.(2023) found that fine-tuning alpaca with 9k samples performs better than the one with 52k data[1].
> Similarly, other work also found that more data (Nx) does not necessarily bring better performance when fine-tuning on multilingual instructions, and even the model trained with down-sampled multilingual samples(1x) obtains better robustness[2].
>
> [1] Chen, Lichang, et al. "Alpagasus: Training a better alpaca with fewer data." arXiv:2307.08701 (2023).
>
> [2] Chen, Pinzhen, et al. "Monolingual or Multilingual Instruction Tuning: Which Makes a Better Alpaca." arXiv:2309.08958 (2023).

---

> > ### Author Response · Authors · 2023-11-16
> > **Response to Reviewer 1odi (2/2)**
> >
> > > Weakness 3:Insufficient literature review in the related work section.
> >
> > Thank you for reminding us! Due to the limitation of space, we only listed some representative work in the related work. We will add more related work in the next version, including InfoXLM[3].
> >
> > [3] Chi, Zewen, et al. "InfoXLM: An Information-Theoretic Framework for Cross-Lingual Language Model Pre-Training." NAACL 2021.

---

> ### Comment · Reviewer_1odi · 2023-11-22
>
> Thanks for addressing my comments on the ablation experiments of CIT.
>
> However, I would note that the BLOOM with AFP and BLOOMZ in Table 1 and Table 3 are not directly comparable because the AFP is under a bilingual setup but BLOOMZ is multilingual. The setup in Table 7 is fair for BLOOM w/ AFP vs BLOOMZ, and the proposed methods slightly outperform BLOOMZ while using much fewer data as mentioned in your response.
>
> Although the combination of contrastive learning and instruction tuning is simple, "using multilingual contrastive learning alone cannot enhance the cross-lingual ability" is a good point.
>
> This clarifies some points for me, and I have updated the rating.
>
> Some suggestions:
> - Highlight the finding: "using multilingual contrastive learning alone cannot enhance the cross-lingual ability".
> - Clarify the experimental setups, and distinguish the bilingual and multilingual setups.

---

> > ### Author Response · Authors · 2023-11-23
> >
> > Thanks for your constructive suggestions! We will incorporate them in the next version of our paper.